# Slow light topological photonics with counter-propagating waves and its active control on a chip

Abhishek Kumar[1,2,4], Yi Ji Tan[1,2,4], Nikhil Navaratna[1,2], Manoj Gupta[1,2], Prakash Pitchappa[3] & Ranjan Singh[1,2]✉

Topological slow light exhibits potential to achieve stopped light by virtue of its widely known robust and non-reciprocal behaviours. Conventional approach for achieving topological slow light often involves flat-band engineering without disentangling the underlying physical mechanism. Here, we unveil the presence of counter-propagating waves within valley kink states as the distinctive hallmark of the slow light topological photonic waveguides. These counter-propagating waves, supported by topological vortices along glide-symmetric interface, provide significant flexibility for controlling the slowness of light. We tune the group velocity of light by changing the spatial separation between vortices adjacent to the glide-symmetric interface. We also dynamically control the group delay by introducing a non-Hermitian defect using photoexcitation to adjust the relative strength of the counter-propagating waves. This study introduces active slow light topological photonic device on a silicon chip, opening new horizons for topological photon transport through defects, topological light-matter interactions, nonlinear topological photonics, and topological quantum photonics.

Slow light effect in photonic systems is of fundamental importance for science and technology, as it offers enhanced light-matter interactions and could significantly miniaturize photonic devices. Efficient control of slow light can lead to various optical applications, such as switching, buffering and modulation, while advancing our understanding of wave physics. Driven by these motivations, efforts have been made to realize slow light effects using conventional photonic crystal (PC) waveguides[1,2], Brillouin scattering[3] and coupled-resonator waveguides[4,5]. Notably, the reduction in group velocity enhances the light-matter interaction, making slow light modes more susceptible to backscattering in the presence of sharp corners and fabrication imperfections. Consequently, this possesses severe constraint for the slow light-based photonic devices for on-chip and integrated photonic applications. Photonic topological insulator (PTI)[6,7] offers an evident

way to resolve backscattering and bending losses induced by disorder and sharp corners[8,9]. The interface of two topologically distinct domains supports robust transport of light with negligible bending and backscattering losses.

Such robustness is derived from the nontrivial bulk topology. In particular, topological states that exist at the interface exhibit linear dispersion throughout the entire band gap and do not usually behave as slow light due to the lack of flat band regions. Although, some proposals have been made to realize the topological slow light modes[8–10]. However, most of them require complex structures[11,12] (e.g., multilayers) or gyromagnetic materials[8,10,13], challenging the feasibility of their implementation as on-chip integrated devices. Among various PTIs, the recently discovered valley photonic crystals[14,15] (VPCs) present an efficient way to realize nontrivial topological phase of light.

[1]Division of Physics and Applied Physics, School of Physical and Mathematical Sciences, Nanyang Technological University, Singapore 637371, Singapore. [2]Centre for Disruptive Photonic Technologies, The Photonics Institute, Nanyang Technological University, Singapore 639798, Singapore. [3]Institute of Microelectronics, Agency for Science, Technology and Research, 2 Fusionopolis Way, Singapore 138634, Singapore. [4]These authors contributed equally: Abhishek Kumar, Yi Ji Tan. ✉e-mail: ranjans@ntu.edu.sg

This is achieved by breaking spatial-inversion symmetry, leading to the emergence of contrasting Berry curvature profiles near the high symmetry corners of Brillouin zone (i.e., K and K′ valleys) of a hexagonal lattice. Notably, the robust transport in VPC relies fully on geometric symmetries. As a result of this, VPC waveguide with sharp corners including 60-degree or 120-degree bends which preserve the hexagonal lattice symmetry exhibits negligible bending loss[14]. However, quantifying the topological protection of transport of light in VPC is still an active area of research[16–19].

Besides, VPC structures can be implemented in compact on-chip devices with periodicity on the order of the wavelength, making them highly suitable for practical applications. More recently, the slow light effect has been shown in bearded interface VPC waveguide[20]. However, the underlying mechanism that could explain the physical picture for the emergence of slow light effect in VPC waveguide is unclear.

In this study, we decipher the realm of slow light topological photonics by observing counter-propagating waves in VPC waveguides. These counter-propagating waves emerge due to the interplay between magnetic phase vortices and a glide-symmetric bearded interface. The interaction between forward and counter-propagating waves leads to the slow light effect in VPC waveguides. In addition to uncovering this phenomenon, we have developed a unique way to tune the slowness by engineering the interface and band gap of the VPC waveguide. Moreover, we demonstrate an active control of topological slow light by photoexciting the interface of the VPC waveguide. We achieved active tuning of group delay (GD) by introducing a local non-Hermitian defect by photoexcitation. Compared to conventional PC platforms, the vortex-driven slow light effect in VPCs offer flexibility to control the slowness by engineering the interaction between forward and counter-propagating waves. To highlight the uniqueness and salient features of our work, we present comprehensive tables (Table 1) that compares the experimental performance specifications of VPC waveguide with existing state-of-the-art photonic platforms. Our work lays the groundwork for feasible integration of the topological slow light effect into photonic platforms[21,22], thereby enriching diverse photonic functionalities and pushing forward the frontier of topological light-matter interaction[23,24].

## Results and discussion
### Slow light topological photonics

In this section, we enumerate the underlying physics of slow light topological photonics. We consider a VPC structure consisting of equilateral triangular holes arranged in a hexagonal lattice with periodicity $a = 260$ μm. The unit cell of the VPC contains two equilateral triangular air holes with side lengths $l_1$ and $l_2$ (Fig. 1a). When the side length of triangular holes is equal, $l_1 = l_2$, the VPC hosts a pair of degenerated Dirac points at K and K′ valleys (see Supplementary Section S1). The introduction of geometrical asymmetricity by setting $l_1 \neq l_2$ breaks the spatial-inversion symmetry which lifts the degeneracy of Dirac points and opens a photonic band gap (see Supplementary Section S1). Consequently, VPC acquires a nontrivial topological phase, resulting in a complementary Berry curvature distribution near the K and K′ Brillouin zone corners (see Supplementary Section S1). To design the VPC waveguide, we create two distinct unit cells, Type A and Type B, each containing two non-equivalent equilateral triangular air holes with side length: $l_1 = \frac{0.7a}{\sqrt{3}}$ and $l_2 = \frac{1.3a}{\sqrt{3}}$ for Type A, $l_1 = \frac{1.3a}{\sqrt{3}}$ and $l_2 = \frac{0.7a}{\sqrt{3}}$ for Type B. For transverse electric (TE) modes where the electric field lies within the plane (xy plane), the out-of-plane magnetic field ($H_z$) component represents the Floquet eigenstate of the photonic crystal. Inspecting the Floquet eigenstates of the VPC unit cell near the K and K′ valleys reveals a self-rotating vortex phase profile. Using the plane wave expansion (PWE) method (see Methods), we calculated the magnetic phase vortex at K and K′ valley for the lower frequency band (Fig. 1b), showing a full 2π rotation centered around the smaller triangular air hole. Here, as highlighted in Fig. 1b, the phase increases in the anti-clockwise direction for Type A unit cell and clockwise direction for Type B unit cell, analogous to valley dependent orbital magnetic moments of electrons.

For the TE modes, the Poynting vector (**S**) of the electromagnetic waves is defined as $\mathbf{S} = \mathbf{E}_\parallel \times \mathbf{H}_\perp$, where $\mathbf{E}_\parallel$ and $\mathbf{H}_\perp$ are the in-plane electric field and out-of-plane magnetic field components, respectively. The magnetic phase vortex imparts a vortex characteristic to the Poynting vector in the unit cells, with reversed chirality for Type A and Type B unit cells (see Fig. 1a, b). Moreover, for a given valley, the chirality of the Poynting vector is opposite for Type A and Type B unit cells (see Fig. 1a). Interfacing both Type A and Type B unit cells in the construction of VPC waveguide domain introduces a sharp kink in the order parameter[25]. Consequently, the topological modes at the domain wall of VPC waveguide are commonly referred as kink states, which exhibits valley-dependent chirality.

VPC waveguides can be designed by carefully arranging the unit cells to form either a zigzag or a bearded interface. The different spatial configurations of magnetic phase vortices at these interfaces result in contrasting transport behaviors of kink states. For instance, zigzag interface with mirror symmetry yields only forward propagating kink states, whereas bearded interfaces with glide symmetry (reflection + translation) permit the coexistence of both forward and counter-propagating waves (see Supplementary Movie 1 and 2). Noteworthy, for the appearance of counter-propagating waves in VPC waveguide, two conditions need to be satisfied: (a) the existence of magnetic phase vortices with non-zero Berry curvature, and (b) the stacking of unit cells with a bearded interface with glide symmetry.

**Table 1 | Comparison of our topological slow light study with existing state-of-the-art photonic platforms**

| Platforms | Conceptual advancement | Active control | Robust transport at sharp bends | Group index ($n_g$) |
|---|---|---|---|---|
| W1 PC waveguide[36] | Waveguide engineering | No | No | 34 |
| PC waveguide[37] | Waveguide engineering | No | No | 90 |
| PC waveguide[38] | Waveguide engineering | No | No | 100 |
| PC waveguide[39] | Waveguide engineering | Yes | No | 100 |
| Magneto-optical photonic crystal[8] | N.A. | No | Yes | 13.26 |
| Coupled resonators[40] | Flat bands via resonators coupling | No | Yes | 33.6 |
| VPC (membrane)[20] | N.A. | No | Yes | 37 |
| VPC (silicon rods)[17] | N.A. | No | Yes | 300 |
| VPC (membrane) | **Counter-propagating slow light** | **Yes** | **Yes** | **39[a]** |

Text in bold – Our work in this article.

*PC* Photonic crystal, *VPC* Valley photonic crystal.

[a]Group index ($n_g$) is calculated for no pump case by using the formula $n_g = c/v_g$, where c is the speed of light in vacuum, $v_g$ is the group velocity, extracted using $v_g = L/GD$. Here GD is the group delay at slow light frequency and L is the length of device equal to 8.97 mm.

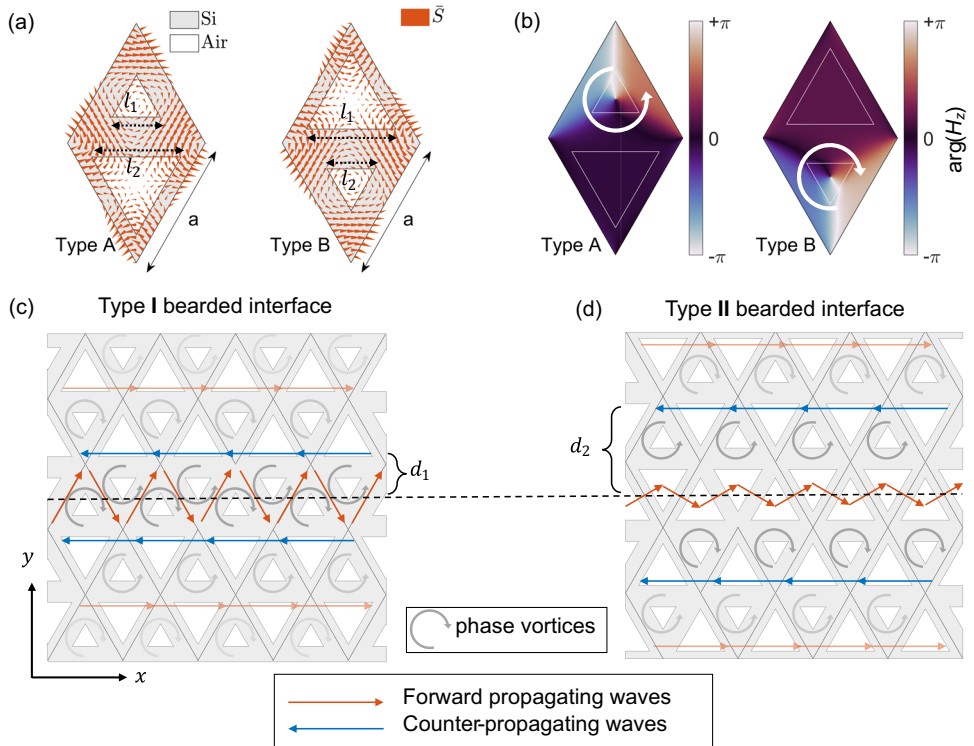

**Fig. 1 | Magnetic phase vortex enabled chiral, robust, and slow light propagation in VPC waveguide. a** Diamond-shaped unit cells of VPC with lattice constant $a = 260\,\mu m$. Type A and Type B unit cells are mirror symmetric. Gray and white portions represent silicon (Si) and air, respectively. Orange arrows represent the locus of the Poynting vector (**S**) at K valley of the lower frequency TE band, exhibiting opposite chirality in Type A and Type B unit cells. **b** The phase distribution of the out-of-plane magnetic field ($H_z$) at K valley of the lower frequency TE band, showing opposite magnetic phase vortices for Type A and Type B cell unit cells. **c**, **d** VPC waveguide with Type I and Type II bearded interfaces. Type I bearded interface comprises smaller triangular air holes, while Type II bearded interface consists of larger triangular air holes at the interface. Orange and blue arrows highlight the forward and backward propagating components of kink states, respectively. In comparison to Type II VPC waveguide, the counter-propagating waves (blue arrow) exist closer ($d_1 < d_2$) to the domain wall in Type I VPC waveguide.

To demonstrate the significance of both these conditions for the existence of counter-propagating waves in the VPC, we have included Supplementary Movie 3. This movie demonstrates the evolution of the Poynting vector profile, transitioning from having only forward propagating waves to both forward and counter-propagating waves in the bearded interface VPC waveguide as we sweep across the frequencies corresponding to wavevectors from the projected K-valley to the band edge. As the magnetic phase vortices exist for wavevectors ranging from the projected K point ($k_x = 2\pi/3a$) to the band edge ($k_x = \pi/a$), we observe the appearance of counter-propagating waves from the projected K point frequency to the band edge frequency, as clearly evident in Supplementary Movie 3.

It is worth emphasizing that the spatial location of counter-propagating waves can be altered by changing the type of bearded interface. For example, in Type I bearded interface with adjacent smaller triangular holes, the magnetic phase vortex singularity lies closer (along the y-direction) to the domain wall, allowing counter-propagating waves to exist closer ($d_1$) to the interface. This starkly contrasts with bearded interface made using neighboring larger triangular air holes, labeled as Type II in Fig. 1c, where the magnetic phase vortex singularity being farther away from the interface allows the counter-propagating waves to exist farther vertically ($d_2$) from the interface. In both Type I and Type II bearded interfaces, the interaction between forward propagating and adjacent counter-propagating waves give rise to slow light effects in VPC waveguides. In fact, the counter-propagating waves temporarily trap the kink states within each magnetic phase vortex before allowing propagation in the direction of the forward kink states, resulting in a low group velocity (see Supplementary Movie 2).

## Passive and active tuning of topological slow light

The emergence of slow light effect at bearded interfaced VPC waveguide can be seen by the flat band dispersion in the projected band diagram (at $k_x = \pi/a$), as shown in Fig. 2. To quantify the strength of slow light, we analyze the group velocity ($v_g$) of valley kink state, which can be adjusted by varying the coupling strength between forward and counter-propagating waves. We achieve this by engineering the interface of the VPC waveguide.

Recent research by Yi Ji et al.[26] have demonstrated that the bandwidth of the topological kink states in bearded interface VPC waveguide can be significantly controlled by altering the shape and size of the air holes in a VPC unit cell. By increasing the band gap of the VPC, the spatial decay rate of the transversely localized electromagnetic fields of valley kink states into the bulk can be enhanced, providing an unprecedented way to tune the coupling strength between the forward and counter-propagating waves. Consequently, we design several VPC waveguides by varying the shape and size of the air holes, as shown in Fig. 2. The shape of the air holes is considered as equilateral polygons with number of vertices ($v$) expressed in multiples of 3 to preserve the $C_3$ rotational symmetry.

The projected band diagrams of TE modes for various bearded interface VPC waveguide (Type I and Type II) incorporating air holes with $v = 3, 6$ and 9 are analyzed using the PWE method (see Methods). The calculated projected band diagrams are shown in Fig. 2a–c with insets showing the corresponding unit cells. The unit cell has a lattice constant $a = 260\,\mu m$ with air holes of side lengths ($l_1 = \frac{1.3}{\sqrt{3}}a, l_2 = \frac{0.7}{\sqrt{3}}a$), and radii ($r_1 = \frac{a}{3}, r_2 = \frac{a}{12}$), ($r_1 = \frac{a}{\sqrt{6}}, r_2 = 0$) for $v = 3, 6$, and 9, respectively. In this manuscript, side length $l$ are defined in proportion to the radius $r$ (centroid to vertex) of equilateral triangles, given by $l = r\sqrt{3}$. In

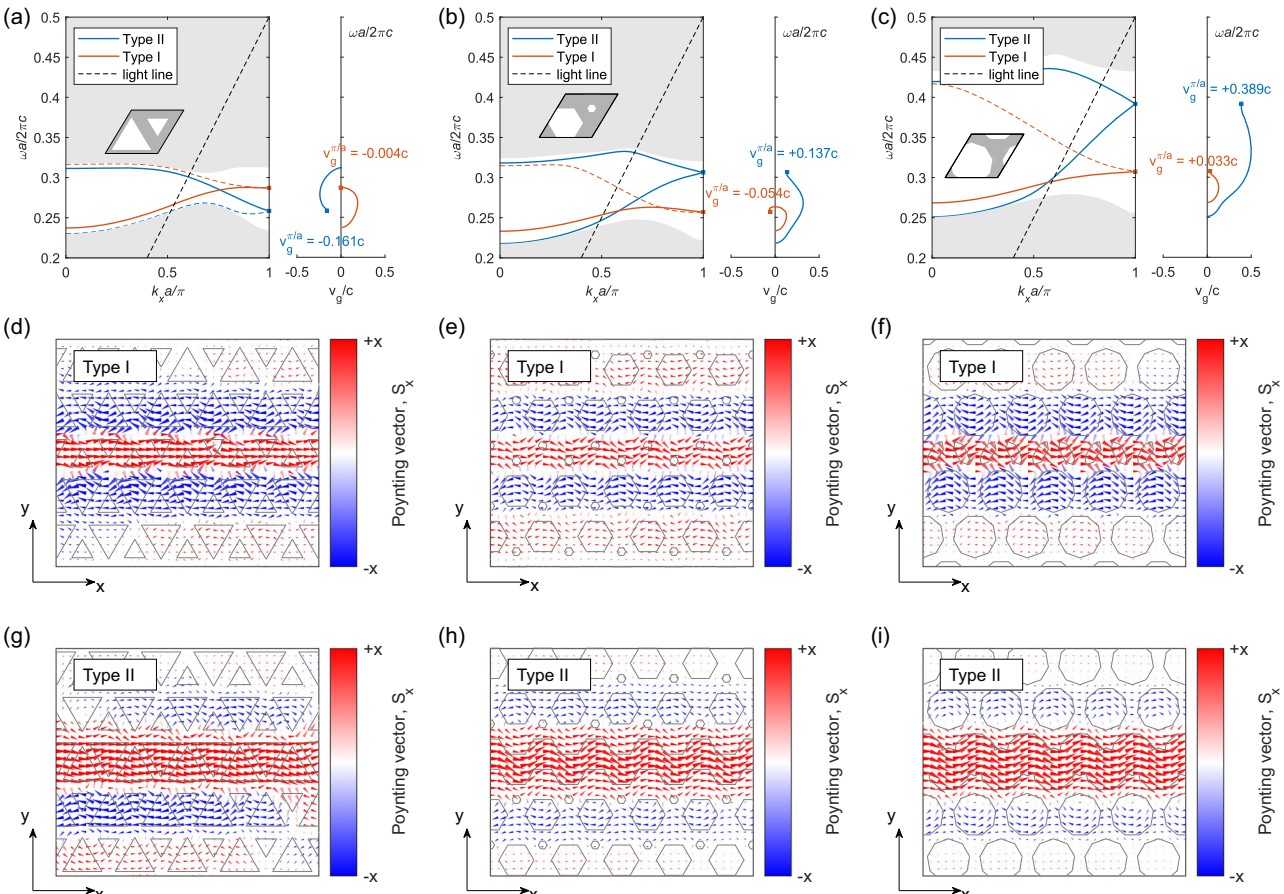

**Fig. 2 | Tuning of slow light effect in VPC waveguide by interfacial engineering.** Projected band diagram and group velocity ($v_g$) for Type I and Type II bearded interface VPC waveguide for (**a**) triangular holes, $\nu = 3$, (**b**) hexagonal holes, $\nu = 6$ and (**c**) nonagonal holes, $\nu = 9$. Solid and dashed line represent the topological and trivial mode, respectively. The right inset in **a**–**c** show the $v_g$ of kink state for $\nu = 3, 6$ and 9, respectively. The square points mark the group velocity at band edge (i.e., $v_g^{\pi/a}$). For Type I bearded interface, $v_g^{\pi/a}$ (red square marker) remains relatively unchanged for $\nu = 3,6$ and 9, while it increases for Type II bearded waveguide (blue square marker for lower topological band) upon increasing the band gap by varying

$\nu$ from 6 to 9. **d**–**f** Contour of Poynting vector flow in Type I bearded interface VPC waveguide for triangular, hexagonal and nonagonal holes, respectively. The red and blue arrows depict the Poynting vectors of forward propagating (+x direction) and adjacent counter-propagating (−x direction) waves, respectively. **g**–**i** Contour of Poynting vector flow in Type II bearded interface VPC waveguide for triangular, hexagonal and nonagonal holes, respectively. The red and blue arrows depict the Poynting vectors of forward propagating (+x direction) and adjacent counter-propagating (−x direction) waves, respectively.

Fig. 2a−c, the gray region represents the bulk bands, while the orange and blue lines depict the dispersion of valley kink states for Type I and Type II bearded interfaces, respectively. It is observed that both these interfaces support two confined modes (highlighted by solid and dash lines in Fig. 2a−c), of which one corresponds to the topological kink state (solid line) while the other represents a trivial state (dashed line)[20,27]. The degeneracy of these two modes at the band edge ($k_x = \pi/a$) is forced by the glide plane symmetry of the interface. It is noteworthy that, to precisely distinguish between the topological and trivial modes within the bearded interface VPC waveguide, we assessed the transmission spectra through Z-shaped waveguide (see detailed discussion in Supplementary Section S1).

Examining the change in the band gap of the VPC waveguide by varying $\nu$ from 3 to 9, we investigate the effect of band gap on the effective group velocity ($v_g$) of the kink state. The $v_g$ for kink state at band edge ($v_g^{\pi/a}$ where $k_x = \pi/a$) for Type I and Type II bearded interfaces is shown in the right inset of Fig. 2a−c, marked by square dots. In the case of Type I bearded interface, $v_g^{\pi/a}$ remains relatively unchanged with an increase in the band gap due to closer proximity of the counter-propagating waves to the interface (Fig. 1c). However, for Type II interface, increasing the band gap increases the transverse decay of the kink state perpendicular to the interface, resulting into

weak coupling between the forward and counter-propagating waves which causes $v_g^{\pi/a}$ to be large for $\nu = 9$ (Fig. 2c).

This observation is further justified through the Poynting vector plot in Fig. 2d−i. The locus of Poynting vector flow is simulated in the frequency domain using COMSOL Multiphysics at the band edge frequency. The red and blue arrows highlight the Poynting vector flow of forward and counter-propagating waves, respectively. As explained earlier, due to the closer proximity of counter-propagating waves to the Type I bearded interface, their intensity does not change significantly with an increase in band gap, depicted in Fig. 2d−f. While for Type II bearded interface, increasing the band gap reduces the spatial extent of the kink states (see Fig. 2g−i), leading to weak coupling between the forward and counter-propagating waves.

To further emphasize the effect of the band gap on $v_g$, we conducted an extensive investigation by fixing the air hole shape to be a hexagon ($\nu = 6$) and varied the air hole radii (centroid to vertex) for a unit cell with period $a = 260\,\mu m$. Figure 3a presents the band gap of VPCs, with the colormap representing the strength of the band gap. We selected the unit cells with increasing band gap, as marked in Fig. 3a. The specific radii ($r_1, r_2$) of these chosen unit cells are as follows: $\left(\frac{0.37}{\sqrt{3}}a, \frac{0.67}{\sqrt{3}}a\right)$, $\left(\frac{0.33}{\sqrt{3}}a, \frac{0.71}{\sqrt{3}}a\right)$, $\left(\frac{0.29}{\sqrt{3}}a, \frac{0.75}{\sqrt{3}}a\right)$, $\left(\frac{0.22}{\sqrt{3}}a, \frac{0.78}{\sqrt{3}}a\right)$ and

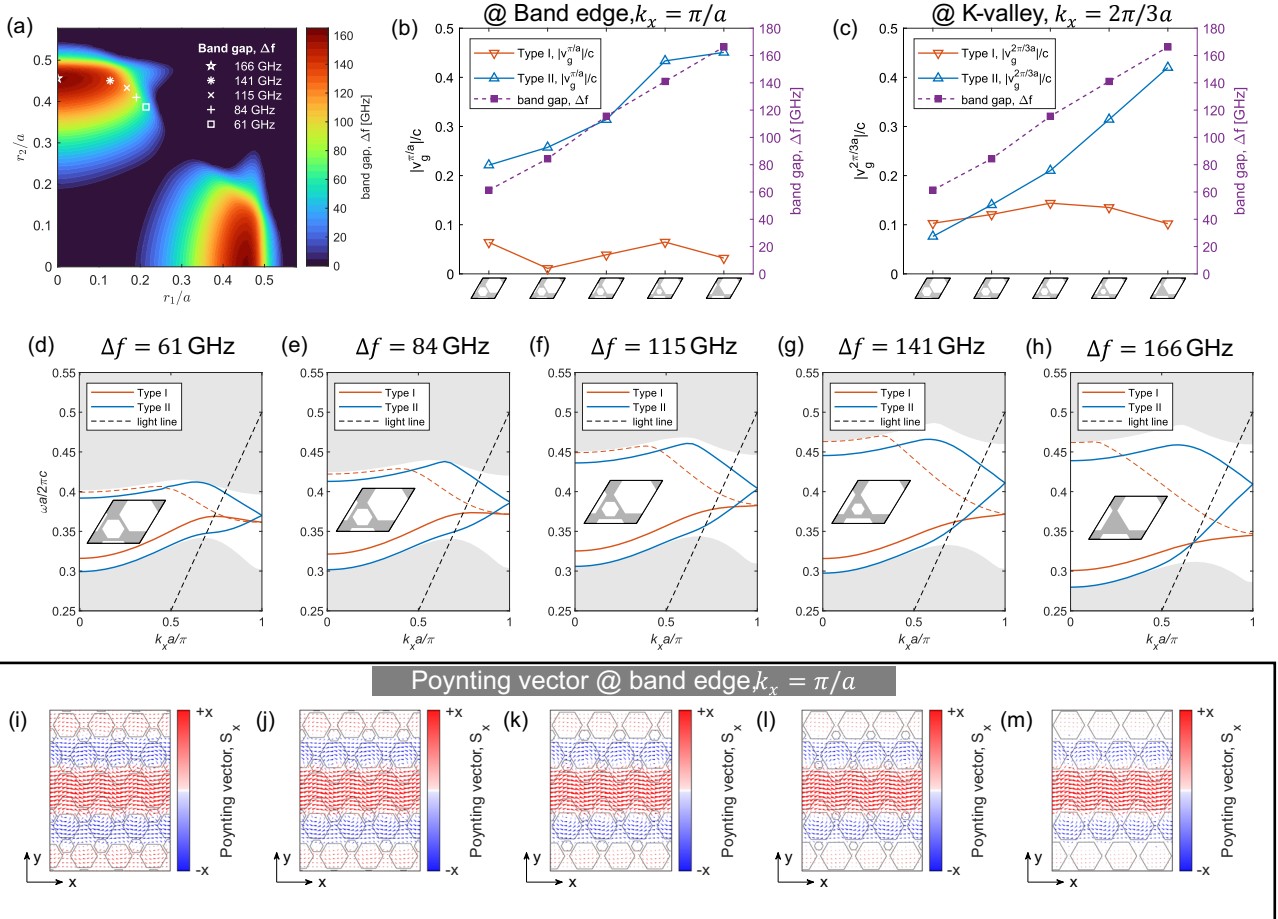

**Fig. 3 | Tuning the slowness of VPC through band gap engineering. a** The band gap of VPCs as a function of air hole radii ($r_1, r_2$) for hexagonal air holes ($\nu = 6$) of period $a = 260\,\mu m$. The colormap indicates the strength of band gap, while markers represent the selected unit cells with increasing band gap. The specific radii ($r_1, r_2$) of the chosen unit cells are as follows: $\left(\frac{0.37}{\sqrt{3}}a, \frac{0.67}{\sqrt{3}}a\right)$, $\left(\frac{0.33}{\sqrt{3}}a, \frac{0.71}{\sqrt{3}}a\right)$, $\left(\frac{0.29}{\sqrt{3}}a, \frac{0.75}{\sqrt{3}}a\right)$, $\left(\frac{0.22}{\sqrt{3}}a, \frac{0.78}{\sqrt{3}}a\right)$ and $\left(0, \frac{0.79}{\sqrt{3}}a\right)$, with corresponding band gap ($\Delta f$) values 61 GHz, 84 GHz, 115 GHz, 141 GHz and 166 GHz, respectively. **b** Band edge group velocity ($v_g^{\pi/a}$) for Type I (orange solid line) and Type II (lower blue solid line) bearded interface VPC waveguides, shown on the left Y-axis. The band gap corresponding to the selected unit cells is shown on the right Y-axis (purple line). **c** Group velocity at projected K-value (i.e., $k_x = 2\pi/3a$) for Type I and Type II bearded interface VPC waveguides, shown on the left Y-axis. Similarly, the band gap corresponding to the selected unit cells is shown on the right Y-axis (purple line). **d–h** The projected band diagrams corresponding to the chosen unit cells, marked in **a**. The band gap increases from 61 GHz to 166 GHz from left to right. **i–m** The locus of the Poynting vector is extracted at band edge i.e., $k_x = \pi/a$. The unit cell of the corresponding VPC waveguide is shown in the inset of **d–h**.

$\left(0, \frac{0.79}{\sqrt{3}}a\right)$, with corresponding band gap ($\Delta f$) of 61 GHz, 84 GHz, 115 GHz, 141 GHz, and 166 GHz, respectively. The projected band diagrams for these chosen unit cells are depicted in Fig. 3d–h, showing a clear increase in the band gap from left to right. Figure 3b shows the group velocity extracted at band edge ($v_g^{\pi/a}$ where $k_x = \pi/a$) for Type I and Type II bearded interface VPC waveguides. For Type I bearded interface, the proximity of the counter-propagating wave to the interface causes the slowness (or $v_g^{\pi/a}$) to remain relatively unchanged, even with an increase in band gap, as shown by orange solid line in Fig. 3b. In contrast, for Type II bearded interface, increasing the band gap reduces the spatial extent of the topological modes perpendicular to the interface, leading to weak coupling between the forward and adjacent counter-propagating waves. Consequently, $v_g^{\pi/a}$ increases with the increase in band gap as shown by blue solid line in Fig. 3b. This observation is further supported by the Poynting vector (**S**) plots depicted in Fig. 3i–m, where we observe gradual decrease in the strength of the counter-propagating waves with an increase in band gap (from left to right) for Type II interface, which results in an increase in $v_g^{\pi/a}$.

Moreover, to demonstrate the universality of the underlying mechanism of slow light in bearded interface VPC waveguide, which arises from the interaction between forward and counter-propagating waves, we calculated the group velocity at the projected K valley (i.e., $v_g^{2\pi/3a}$ where $k_x = 2\pi/3a$), as shown in Fig. 3c. Similarly, there is no correlation between $v_g^{2\pi/3a}$ and the band gap for Type I bearded interface while $v_g^{2\pi/3a}$ increases with the band gap for Type II interface, which is also supported by the Poynting vector plots (Supplementary Fig. 5).

In addition to interfacial engineering, we experimentally demonstrate an active control to the slow light effect in the VPC waveguide through photoexcitation. To showcase this, we fabricated VPC waveguide chip of Type I bearded interface and $\nu = 3$ operating in the terahertz frequency band. The Type I bearded interface was chosen for the active control demonstration since the passive tuning of the kink state group velocity through band gap engineering is unfeasible, as shown in Fig. 3b and 3c. The VPC waveguide was fabricated on a high-resistivity silicon (HR-Si) wafer with a resistivity of 10 kΩ·cm. The HR-Si offers large refractive index contrast, low absorption loss and ensures CMOS compatibility. The geometrical parameters of the fabricated bearded VPC waveguide chip are as follows: periodicity $a = 260\,\mu m$,

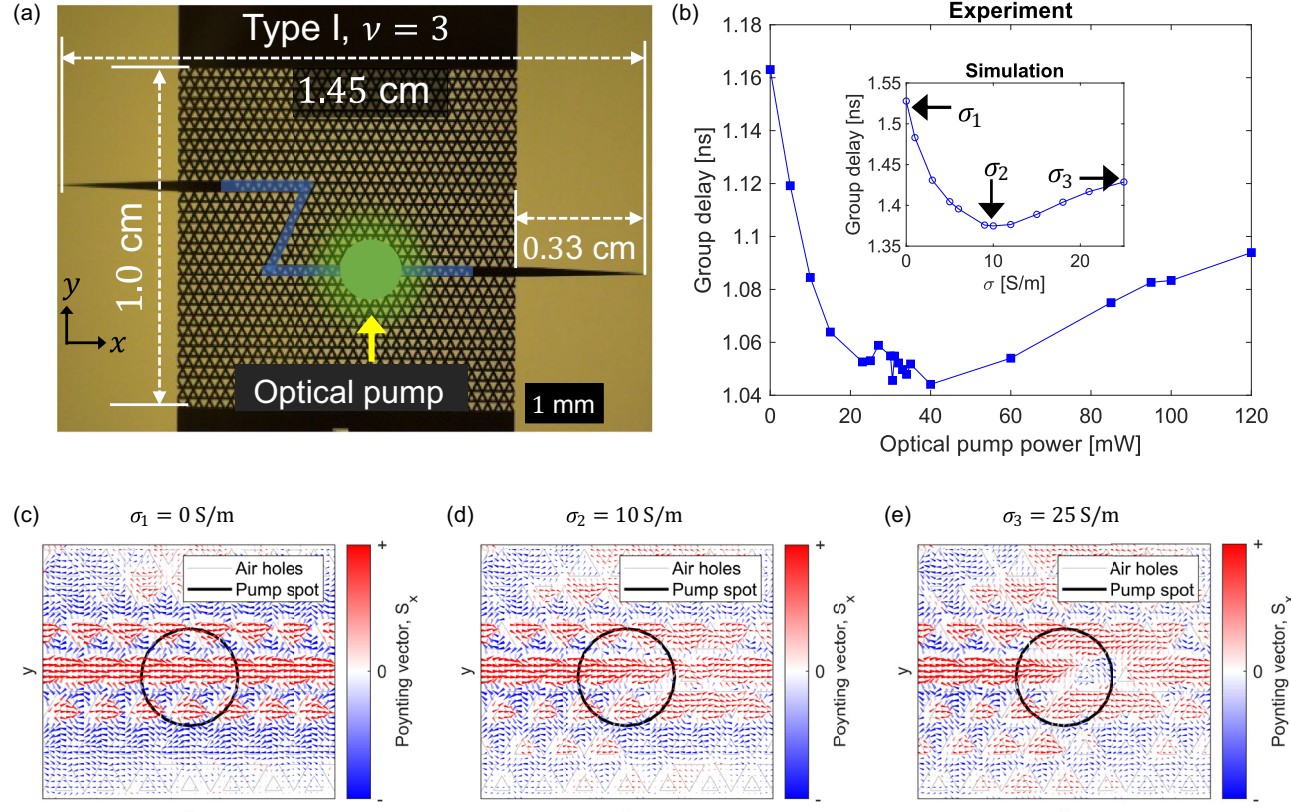

**Fig. 4 | Active slow light topological waveguide. a** Optical image of the fabricated Type I bearded interface VPC chip with $\nu = 3$. The domain wall of the VPC waveguide is highlighted by blue shaded regions. Adiabatic input and output couplers are attached to couple the terahertz waves in and out of the VPC waveguide chip. All critical dimensions are highlighted. The green circular region indicates the location of the optical pump spot ($\lambda = 532$ nm). **b** The experimental GD is plotted as a function of optical pump power. The GD values are obtained using a vector network analyzer (VNA) based setup. The inset graph illustrates the simulated GD against the conductivity ($\sigma$) of silicon (Si), which shows good agreement with experimental results. For the simulation, a 3D approach is employed in CST Microwave Studio to replicate the experimental conditions. **c–e** The locus of Poynting vectors is calculated at different conductivity of Si, as marked in the inset of graph (**b**). The black circle encloses the region with variable Si conductivity.

$l_1 = \frac{0.7}{\sqrt{3}}a$ and $l_2 = \frac{1.3}{\sqrt{3}}a$ for Type A and $l_1 = \frac{1.3}{\sqrt{3}}a$ and $l_2 = \frac{0.7}{\sqrt{3}}a$ for Type B unit cells. The thickness of the Si membrane is kept at 200 µm to ensure good vertical confinement through total internal reflection. Fig. 4a depicts the optical image of the fabricated VPC waveguide chip, where the bearded interface is highlighted by a blue shaded region. As noted in Fig. 2a, two confined modes exist for a bearded interface, corresponding to the topological and trivial states. Building upon the ability of kink states to overcome sharp bends[15,20,27,28], we eliminate the contribution of trivial states in the transmission spectrum by constructing Z-shaped domain walls featuring two sharp bends.

As discussed above, the presence of counter-propagating waves in the vicinity of the bearded interface facilitates the slow light effect in VPC waveguide. Therefore, we sought to actively control the spatial distribution of these counter-propagating waves through photo-excitation, providing a way to tune the slow light effect in VPC waveguide. It is worth noting that, akin to tuning the slowness of kink state through band gap engineering (as shown in Figs. 2 and 3), photo-excitation also modifies the relative strength of the counter-propagating waves by modulating the terahertz waves through free carriers. To experimentally quantify the slowness of kink states, we measure the group delay (GD), as it directly quantifies the time delay experienced by signal or electromagnetic waves as they propagate through the waveguide.

We record the GD defined as GD $= -\frac{d\phi}{d\omega}$ (where $\phi$ and $\omega$ are phase and angular frequency of the transmitted wave, respectively) from the VPC waveguide chip using a vector network analyser (VNA) based experimental setup (see Supplementary Section S5). We photoexcite the domain wall of VPC waveguide chip (see Fig. 4a) with a continuous wave laser of wavelength 532 nm. The optical beam spot (diameter ≈ 0.5 mm) (see Supplementary Fig. 7) was kept small to ensure uniform illumination of few unit cells. Fig. 4b shows the variation of GD as a function of optical pump power. To accurately replicate the experimental conditions, we performed 3D simulation in CST Microwave Studio, as thoroughly explained in SI, section S7. Given that the optical beam spot in the experiment is focused on only a few unit cells (Fig. 4a), we emulate the effects of optical pumping by defining a circular region spanning a few unit cells (highlighted in red in Supplementary Fig. 10a) at the center of the bearded interface of the VPC waveguide. Within this region, the conductivity of Si is varied, which is equivalent to changing the optical pump power in experiment. Fig. 4b shows a comparison between the experimental and simulated GD curves, which shows good agreement in the variation of GD. Importantly, we observe an identical relative change (i.e., difference between maximum and minimum GD) in GD values for both the experiment and simulation, which is around 11%. The slight variations in GD values can be attributed to slight experimental misalignment and variations in the thickness (200 ± 25 µm) of the fabricated VPC waveguide.

To understand the behavior of GD as a function of optical pump power (or Si conductivity in simulation), we computed Poynting vector plots from 3D simulations. Figure 4c–e shows the locus of Poynting vector calculated at the conductivity values marked in the inset of Fig. 4b, where the black circle encloses the region with variable Si conductivity. For $\sigma = 0$ S/m, the transport of kink state is accompanied by forward and counter-propagating waves (Fig. 4c) where the

electromagnetic waves are trapped temporally at each vortex before propagating to the next (see Supplementary Movie 2), which is the physical mechanism of topological slow light. As the conductivity of Si increases, free carrier densities increase and imparts losses, causing the photoactive region to act as a non-Hermitian defect that attenuates mainly the backward propagating waves (see Fig. 4d where the strength of backward Poynting vectors in blue weakens after the non-Hermitian defect). The attenuation of backward propagating waves leaves only the forward components and the waves are not trapped temporally at each vortex anymore after the photoactive region, resulting in the decrease of GD. When the backward Poynting vector component is completely attenuated, the GD will be at minimum as shown in Fig. 4d and the inset of Fig. 4b for $\sigma = 10$ S/m. With further increase in the conductivity of Si, both forward and backward propagating waves are attenuated about the position slightly after the photoactive region and the kink state now takes a detour as shown in Fig. 4e. This additional detour causes the GD to increase again as evident in the inset of Fig. 4b for conductivity more than 10 S/m. However, the additional group delay arising from such detour is less than those caused by the temporal trapping of counter-propagating waves at each vortex, hence the GD for $\sigma > 10$ S/m is still lower than the unperturbed case where $\sigma = 0$ S/m. To ensure the consistency of our results, we conducted photoexcitation measurements multiple times by gradually increasing and decreasing the optical pump power. The results are depicted in Supplementary Fig. 17, showing no variation in the measured GD values. This observation rules out the possibility of any hysteresis-type behavior and validate the reproducibility in our experiment.

In conclusion, our findings demonstrate a slow light topological photonic waveguide through emergence of counter-propagating waves, resulting from the presence of magnetic phase vortices and a glide symmetric bearded interface. The interaction between forward and counter-propagating waves gives rise to the slow light effect in VPC waveguides, which can be actively and passively controlled by adjusting the relative magnitude of their Poynting vectors. By engineering the interface, we demonstrated tunable slow light in a passive manner, whereas introducing a non-Hermitian defect through photo-excitation allowed active tuning of group delay. These results hold significant implications for the advancement of on-chip topological photonic devices[29–31] including topological light-matter interactions[24] and nonlinear topological photonics[32]. Moreover, the development of topological cavities based on slow light topological photonic states could serve as a crucial building block for quantum integrated circuits[33]. Additionally, extending slow light topological photonic concept to non-reciprocal topological photonic systems hold promise for achieving ultraslow and stopped light photonic devices[34]. As the field of quantum materials[35] continues to thrive with innovative ideas and research, we envision abundant opportunities at both the fundamental and technological levels that could be facilitated by the heterogenous integration of on-chip slow light topological photonic modes with emerging quantum materials.

## Methods
### Sample fabrication
The VPC chip was fabricated on an 8-inch diameter high-resistivity silicon (Si) wafer with resistivity >10 kΩ•cm. The Si wafer was carefully cleaned, followed by the deposition of 4 µm thick silicon-dioxide (SiO$_2$) layer. We used photolithography to define the triangular etch holes and device separation trenches. SiO$_2$ was selectively etched away, exposing the Si surface. To achieve the uniform side wall of holes, deep reactive ion etching of Si was carried out. The photoresist layer and SiO$_2$ layers were then completely removed. Back grinding of Si was carried out until 200 µm thick Si was left.

### Numerical simulation
We used frequency domain simulation in COMSOL Multiphysics to calculate the locus of Poynting vector. The topological mode of VPC waveguide is excited using a chiral source (an electric dipole with dipole moments $E_x = 1$ and $E_y = i$). We used perfectly matched layer (PML) to surround the VPC waveguide to eliminate the unwanted reflections from the boundary of VPC waveguide. Refined meshing is employed for the air holes of VPC waveguides, ensuring more than ten mesh elements between the smallest geometrical feature.

For 3D simulation we employed CST Microwave studio. The 3D structure is drawn in the CST simulation environment. To neglect the effect of unwanted signals we used open and add space boundary conditions. The simulations are performed using time domain solver. The detailed discussion is present in Supplementary Information.

### Experimental setup
Complex S-parameters are measured using Vector Network Analyzer (VNA) to obtain the group delay. The VNA measurement setup features a Keysight N5222B network analyzer which generates signals in the frequency range 10 MHz to 26.5 GHz. These frequencies are then upconverted to the WR-2.2 frequency range using WM-570 VNAX frequency extension modules. The WR-2.2 VNA setup can measure response starting from 320 GHz. The system is then calibrated using a SOLT (Short-Open-Load-Through) waveguide calibration procedures adhering to WR-2.2 standards. Calibration accounts for system non-idealities and enables accurate measurements of VPC chip response. For more details, we direct the reader to the Supplementary Information.

### Plane wave expansion (PWE) method
The detailed discussion on plane wave expansion is given in the Supplementary Section S8.

## Data availability
The Figure data generated in this study have been deposited in the DR-NTU database at https://doi.org/10.21979/N9/WSLIFH. All other data generated in this study are provided in the Supplementary Information.

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

## Acknowledgements

All the authors acknowledge the research funding support from National Research Foundation (NRF) Singapore, Grant No: NRF-CRP23-2019-0005.

## Author contributions

A.K. and Y.J.T. equally contributed to the work. The concept and experiments design were developed by A.K., Y.J.T. and R.S. The concept of counter-propagating waves was proposed by Y.J.T. and analyzed by A.K., Y.J.T. and R.S. Passive tuning of slow-light was formulated by A.K., Y.J.T. and R.S. The active tuning experiment was designed by A.K. and R.S. A.K. and N.N. performed the active tuning measurement. A.K. and Y.J.T. performed the simulation analysis. Y.J.T. performed the intensive numerical analysis to optimize the photonic crystal parameters. A.K. conducted the 2D and 3D simulation. Y.J.T. performed the plane wave expansion analysis. A.K., N.N. and M.G. performed the transmission measurements. A.K., Y.J.T. and R.S. performed the experimental and 3D simulation data analysis. P.P. fabricated the samples. A.K., Y.J.T., and R.S. analyzed all the data. A.K., Y.J.T. and R.S. wrote the paper with inputs from all co-authors. R.S. led the overall project.

## Competing interests

The authors declare no competing interests.
