## [Peer Review File · Nature Communications]

REVIEWER COMMENTS

Reviewer #1 (Remarks to the Author):

Kumar et al. describe their work on slow light control in bearded valley-Hall (VH) waveguides, both passively by tuning the geometry of the interface and actively by light absorption. The manuscript is overall well-written, the results are sound and the topicality, i.e, active control of slow light in topological photonic structures, is of relevance. However, I fail to see how the presented results fulfil the broad impact requirement for publication in Nature Communications and feel like some of the claims on the underlying physical mechanism for slow light that the authors discuss would require further justification (both through simulations and experiments). I will elaborate on this judgment through the points that follow.

1- The authors seem to claim (through the abstract) that unveiling the mechanism for slow light in bearded VH waveguides fosters an understanding of topological protection in the slow-light regime. However, I cannot find any discussion of this aspect in the rest of the manuscript. Recent works, e.g., [C. Rosiek et al, Nature Photonics 17, 386 (2023)] and [E. Verhagen, Proceedings Volume PC12196, Active Photonic Platforms 2022, SPIE 2022; <https://doi.org/10.1117/12.2632855>], have experimentally addressed the question of topological protection of slow light in near-infrared bearded VH waveguides that use the "exact" same unit cell and interface as here. They have found no advantage of the topological mode over the trivial one. Is the situation any different for the frequencies explored here? The authors should, at least, comment on these recent results and discuss what level and against what type of defects they expect topological protection, which calls for a reassessment of lines 43-47. For example, what is the role of the roughness seen in the contour of the holes in the inset of Fig. 3b?

2- The authors describe the physical mechanism behind slow light in the explored waveguides as that of coupling between closely spaced (adjacent) counter-propagating modes, which originate from magnetic phase vortices that have their locus at the small triangle in the VH photonic crystal unit cells. As much as I believe the physical picture is correct, the authors do not provide a sufficiently clear presentation of that physical picture:

- For example, the way in which the magnetic phase/Poynting vector vortices of single unit cells add up to lead to only forward propagating components in a zigzag interface and to both forward and backward propagating components in the bearded interface (Supplementary Sections S2 and S3) could be made more explicit. The reader is likely capable of constructing such a picture from the unit cell Poynting vector, but, and especially given that it is already in the Supplementary Material, it may help to have a succession of images where the progressive effect of adding vortices is observed.

- More importantly, the wording "counter-propagating modes" is used across all the manuscript, including its title, and the slow light effect is understood as coupling of these modes. Nevertheless, these two (or more depending on the transverse decay) modes are never really shown to exist independently.

What I mean by this is that the authors provide no simulations of how these "modes" look like when only the involved magnetic phase vortices are included. I am unaware if this is possible, but I would encourage the authors to try something. The authors could maybe take regions that are forward-propagating or backward-propagating in the final waveguide, cut out that part in some clever way and embed it in a bulk of the same material. Then simulate the different waveguide "modes" and study the resulting waveguide dispersion as the (strong) coupling of these original modes. I understand that this might not work very well since the few-mode decomposition might be a very bad approximation. In any case, if such an approach is not possible, then I think calling these "counter-propagating modes" is misleading and maybe a better wording would be "counter-propagating adjacent wavefronts" or something along those lines.

3 - With the physical picture at hand, the authors describe the effect of geometry and band gap size on the achieved slow light. Using that picture and the transverse decay of the topological modes perpendicular to the interface (which becomes faster as the gap becomes larger and therefore may control the coupling between the counter-propagating "modes"), they seek to explain the observed group velocities. Even if I understand that the C6 symmetry needs to be preserved, which limits the number of explored cases with the approach taken, I think the general conclusions drawn (though probably right) are not justified. Maybe a study for a fixed value of v and varying hole size asymmetry to control the gap would allow a more continuous sweep of the geometry and more well-supported conclusions. Let me elaborate on this: the authors claim that "examining $v_g^{(\pi/a)}$ reveals minimal change for Type I interface with respect to the bandgap, while contrasting alteration is observed for Type II interface, as clearly evident in the right inset of Fig. 2a-2c." While I agree that such value seems more stable for Type I, I find that the conclusion drawn from it, which is that slow light in Type I waveguides is relatively unimpacted by the band gap width because the counter-propagating regions are in close proximity and are therefore less affected by the transverse decay, is done based on only a few band structures and on a single wavevector. I fail to understand in which way the Brillouin zone edge is a representative reciprocal point in this work, and I would rather expect that, if the physical picture holds, it does also relatively well at other reciprocal vectors. Actually, the only wavevector that may carry a particular meaning for these structures is the projection of the K point along the interface direction, somewhere around $0.7\pi/a$, or the region in its vicinity, where topological properties are guaranteed.

4- Trying to use also the same physical picture, the authors study how to control the group velocity by locally pumping (down to the size of the beam spot) some region of the waveguide and, as a consequence, controlling the coupling between counter-propagating "modes". Regarding this experiment and its comparison to simulations, I also have a number of remarks:

- The authors compare the overall evolution of the experimentally evaluated group index with pump power to the simulated values of the group index as a function of the conductivity of silicon. They then claim that "the simulated n_g exhibits large value compared to experiment" and that "it is attributed due to periodic boundary condition used in the simulation, which uniformly changes the conductivity of Si throughout bearded interface waveguiding region, whereas in experiment the photoexcitation of Si is limited to just one- or two-unit cells." While I might understand that the boundary conditions are important, the disparity is not only huge but the value for no pump power and $\sigma = 0$ does not seem to

match at all, which invalidates their argument. I think the local nature of the excitation (experiment) compared to the periodic nature of the excitation (simulation) is not the reason behind this, although it does certainly play a role. One aspect that is critical when dealing with the evaluation of the group index is which method is used. Are the authors taking a numerical derivative or are they using a formula for it (see for example Eq. 10 in [Lalanne et al. ACS Photonics 9, 4 (2019), where the group velocity can be found at a specific wavevector using Lorentz reciprocity]? If they are using a numerical derivative I suggest that they use the formula. The problem with evaluating the group index using a numerical derivative in glide-plane-symmetric waveguides is that, unless the authors have enforced the glide-plane symmetry to the finite-element mesh, the resulting band structure will have a small gap at the degeneracy point and therefore exhibit a diverging group index at the band edge, i.e., a very large group index when evaluated using the derivative. I have personally had this same issue, which I solved by either an ultra-fine mesh or via such formulas. Given that the evaluations are done at $k = \pi/a$, the very large values they report suspiciously indicate this issue. Somehow, the fact that the PWE simulation with a complex refractive index (roughly the same as having a finite conductivity...) gives much better results also indicates the previous. If the authors have, on the contrary, not used a numerical derivative to calculate n_g , then I would encourage them to revisit their COMSOL band structure model and/or give more details than those in Supplementary Section S10.

- If we disregard the disparity of the values of the group index and focus only on the shape of its evolution with excitation, the authors claim that the simulation behaviour is "akin to experiment". However, I feel that such a statement is not true. The only aspect where simulation and experiment match is that the group index increases initially. However, the experiment seems to exhibit a linear growth of n_g , but not the simulation. After that initial growth, in one case there is an extremely fast (step-like) drop and in the other a quite smooth evolution going across a maximum. In addition, the experiment shows that the group index grows again after that drop. I think a wider experimental data set (measurements on many more waveguides), a reproducibility study and an analysis of possible hysteresis (is there any permanent damage?) would clarify what that drop is. The authors attribute it to "the increased photoexcitation at the magnetic phase vortex singularity, destroying the supported adjacent counterpropagating modes". This might be indeed true, but it is only a hypothesis and, given that the actuation mechanism is supposed to build upon the counter-propagating "modes" model, a hypothesis might not be enough. Finding better proof of this would contribute a lot to the coherence of this part of the manuscript. One possibility would be for the authors to elaborate a slightly more advanced band structure model where instead of a single slice with a finite conductivity/extinction coefficient, they take several slices with changes in extinction coefficient that follow a Gaussian excitation spot (for PWE) or a spatially dependent conductivity following a gaussian excitation spot (for COMSOL). In that case, the pump power increase will be better mimicked through simulation.

- Section S9 seems quite interesting although I fail to understand exactly how the simulation is set. Are the authors using an input and output port in COMSOL or is it also a periodic simulation with the laser spot effect at the centre of the supercell? Please, provide more details than what is in the Methods section. Whatever the details, I find this much more relevant than all the band structure simulations the authors have done, since these simulations are much closer to the experiment they perform. What is really occurring in the experiment is not a group index change or "an active control of slow light", but the creation of a lattice-scale defect through which the topological kink state can travel through. Since this would be the actual underlying mechanism, I would suggest this is brought forward to the main text and

compared to what happens for the trivial branch. That comparison, completely lacking in the manuscript, is very relevant, especially if they want to highlight the benefits of using a topological mode in view of recent negative results on their use to fight backscattering from fabrication imperfection.

In addition to the more fundamental issues just discussed, a number of small technical points, presentation issues or less-important remarks should also be addressed:

5- In L40, the wording "extreme sensitivity to disorders accompanied by fabrications" reads strange. Maybe the authors could rephrase it to "extreme sensitivity to disorder resulting from the fabrication process".

6- The rendering of Figs. 1 and 2 is very poor (in general of all figures) and it becomes hard to see some of the features, e.g., the Poynting vector field lines in 1(a). The review process will benefit a lot from getting accurately rendered images, otherwise, a large portion of the information conveyed by figures is just lost.

7- L86: "the" photonic bandgap reads strange, maybe "a" photonic bandgap is more accurate.

8- The PWE simulations are 2D but the structure is inherently 3D, how good is the agreement? In principle, the PWE could be extended to 3D, right?

9- In L167-173, the authors make the distinction between the two modes as topological and trivial. Can they comment on how this distinction is made? One of the geometries ($v = 3$ and Type I interface) is explored experimentally and the use of 120° bends may be used to identify the topological mode in that one, but the authors should describe their generic method to identify which mode is topological or refer to the appropriate reference if existent.

10 - The Supplementary Video SVa will benefit from some more explanation, e.g., labelling in the video itself.

11 - There is probably a sign mistake in the group velocity plot of Fig. 2a, since the value at $k = \pi/a$ is in principle negative. This sign does not change the core message of the manuscript.

12 - In L227, some reference regarding transmission across a Z-shaped domain is needed.

13 - In Fig. S7, can the authors provide a wider span, where the transmission of the trivial band is shown, or they were limited by the VNA range?

14 - It seems to me that a full analysis of the group index curve as a function of frequency could be done the same way that they do the analysis for the maximum ($k = \pi/a$). It could provide a much more complete picture of the effect of photoexcitation and whether the modelling can capture it correctly (the modelling is done as shown by the group index curves in Fig. 3d).

In summary, the work presented by Kumar et al. provides interesting insights into the mechanisms of slow light in valley-Hall waveguides, but I feel that the simulations related to the underlying mechanism should be extended to a wider span of geometries and the conclusions tested for a larger span of the Brillouin Zone to be fully convincing. In addition to that, the link between such a mechanism and the proposed experimental way of controlling the group delay via photoexcitation should be explored in more detail. This being said I am afraid that, even if all these considerations are taken into account, the work would still not be of the relevance that grants publication in Nature Communications.

Reviewer #2 (Remarks to the Author):

In their study, Kumar et al demonstrated slow light in a topological photonic system that resembles the quantum valley Hall effect in the terahertz regime. While there has been a previous demonstration of this concept by Yoshimi et al (Ref. 22 in the manuscript), the authors were able to elucidate the underlying mechanism of the slow light, which is found to be the counter propagating energy flows induced by topological vortices distributed along the interface. Additionally, the authors experimentally demonstrated that the slow factor can be tuned by optical pumping. The overall quality of the work is good. I am happy to recommend its publication in Nature Communications if the following points can be addressed.

1. What does 'kink' state mean? The authors should provide some description in the text.
2. In slow light applications, the bandwidth delay product is an important figure of merit. The authors should provide some analysis on this parameter and compare their results with other works in the table.
3. The topological protection of quantum valley Hall effect is usually quite limited. Although the wave can bend around carefully engineered corners, I am not sure how robust it is against general fabrication imperfections, compared to conventional photonic crystal waveguides.

Reviewer #3 (Remarks to the Author):

The topological slow light effect has been demonstrated in valley photonic crystal (VPC) waveguides with a bearded interface. In the manuscript, the authors explain the mechanism of the formation of slow light modes in such bearded-interface VPC waveguides by considering the spatial distribution of magnetic

phase vortices around the interface. In addition, the authors experimentally demonstrated the active tuning of group index in a bearded-interface VPC waveguide in the THz regime.

Topological slow-light waveguides are attractive, particularly, from the viewpoint of applications to integrated photonics because they can potentially address some problems in conventional slow-light devices, strong backscattering owing to structural disorders, and huge loss at waveguide bends. Given that, the subject discussed in the manuscript is a very much timely topic and would gather interest in the community.

However, I cannot recommend the manuscript for publication in Nature Communications because of the following reasons, at least in its present form.

1) I can naively understand that the backward-propagating component existing closer to the forward-propagating component makes the group velocity of the valley kink state slower. Is the mechanism unique for bearded-interface VPC waveguides? It is well known that “non-topological” PC waveguides with glide-plane symmetry support slow light modes(*). The authors should discuss the difference in the mechanisms between the two kinds of slow-light waveguides. If the proposed mechanism is unique in topological waveguides, it makes the current work impactful significantly.

(*) for example, C. M. Patil et al., “Observation of slow light in glide-symmetric photonic-crystal waveguides,” Opt. Express 30, 12565 (2022).

2) The authors explain that the kink states with large group indexes are formed when the forward and backward propagation modes are coupled strongly. This explanation infers that these are these two modes originally. But they are not identified. Is it appropriate to call them modes?

3) The authors did not provide enough explanation about the assignment of topological guiding mode. In Type I with $v = 3$, where smaller holes are facing the interface, the lower-frequency branch will be a topological mode (according to [27]). On the other hand, when $v= 6$ and 9 , the higher-frequency branch is assigned to a topological band. How did the authors know which branch of the in-gap modes is topological? Why does the change mentioned above happen?

4) The mechanism for the change in the group index under photoexcitation seems to be explained differently from that in the first half of the manuscript. The authors explain that the extra detour of light, not the change in the distance between the forward- and backward-propagating components, causes the change in the group index. It might be my misunderstanding of the author’s intention. But I feel the authors need to explicitly explain whether the tuning mechanism is the same or not with the mechanism discussed in the first half of the manuscript so that the readers will not misunderstand. To directly show that the distance between the forward- and backward-propagating components changes under the photoexcitation, the authors need to show the distributions of the Poynting vector of the eigenmode when non-zero conductivity.

5) I could not understand the relationship between the inset of Fig. 3 c) and Fig. 3 d). In the inset of Fig. 3 c), the group index has a peak at the conductivity of ~ 50 S/m. Does this conductivity correspond to κ of 0.5 in Fig. 3 d)? Why are the group indexes between the two plots different largely?

6) It was unclear to me how the authors estimated the group index under photoexcitation. I suppose they used Eq. (2) with $L = 8.97$ mm. If yes, the estimated group index should be understood as an "effective" or "averaged" one since the change in the group index only occurs within one- or two-unit cells as the authors discussed in LL265-268. Related to this point, why could the authors the group indexes similar to the experimentally obtained values by using the PWE method?

Response Letter to Reviewers

We thank all the reviewers for their constructive comments and inputs on our work. The comments have helped us to enrich the content of the manuscript.

In the subsequent text, the comments from the reviewer are presented in 'black' text, followed by our comprehensive response highlighted in 'blue'. Taking into consideration the reviewer's comment, the revisions incorporated in the revised manuscript and supplementary information are highlighted in yellow. The modified texts in the revised manuscript are posted here in the *italic* font.

Reviewer #1 (Remarks to the Author):

Kumar et al. describe their work on slow light control in bearded valley-Hall (VH) waveguides, both passively by tuning the geometry of the interface and actively by light absorption. The manuscript is overall well-written, the results are sound and the topicality, i.e, active control of slow light in topological photonic structures, is of relevance. However, I fail to see how the presented results fulfil the broad impact requirement for publication in Nature Communications and feel like some of the claims on the underlying physical mechanism for slow light that the authors discuss would require further justification (both through simulations and experiments). I will elaborate on this judgment through the points that follow.

R1.1: The authors seem to claim (through the abstract) that unveiling the mechanism for slow light in bearded VH waveguides fosters an understanding of topological protection in the slow-light regime. However, I cannot find any discussion of this aspect in the rest of the manuscript. Recent works, e.g., [C. Rosiek et al, Nature Photonics 17, 386 (2023)] and [E. Verhagen, Proceedings Volume PC12196, Active Photonic Platforms 2022, SPIE 2022; <https://doi.org/10.1117/12.2632855>], have experimentally addressed the question of topological protection of slow light in near-infrared bearded VH waveguides that use the "exact" same unit cell and interface as here. They have found no advantage of the topological mode over the trivial one. Is the situation any different for the frequencies explored here? The authors should, at least, comment on these recent results and discuss what level and against what type of defects they expect topological protection, which calls for a reassessment of lines 43-47. For example, what is the role of the roughness seen in the contour of the holes in the inset of Fig. 3b?

Our response:

We thank the reviewer for the comment. Here, we want to highlight that in this study, we have elucidated the mechanism behind the topological slow light effect in the bearded interface valley photonic crystal (VPC) waveguide. It is important to emphasize that further investigations specially dedicated to exploring the topological protection of these slow light modes in VPC are imperative, which would greatly benefit from the insights gained through our work of the underlying physical picture of topological slow light modes.

In response to reviewer's suggestion, we undertook a comprehensive examination of the impact of defects on slow light modes within our fabricated VPC waveguide (Type I, $\nu = 3$). We introduced these defects through two distinct approaches i) Firstly, by varying the dimensions of triangular holes and ii) secondly, by removing the triangular holes from the waveguiding path. The corresponding Poynting

vectors associated with these defects are calculated in COMSOL Multiphysics, with detailed procedure elucidated in the subsequent part of the response letter. The results of these simulations are depicted in Fig. R1. In Fig. R1a-R1c, the dimensions of six triangular holes were varied, ranging from $l_1 = 0.4a/\sqrt{3}$ to $l_1 = 0.55a/\sqrt{3}$, where $a = 260 \mu\text{m}$ represents the periodicity of unit cell. Likewise, in Fig. R1d-R1f, we selectively removed one to three triangular holes. Examining the locus of Poynting vectors in Fig. R1a-R1f reveal that the kink state takes a detour around the defect region while retaining its forward and counter-propagating moving waves characteristic even after traversing the region of geometrical imperfections, exhibiting robust transport behaviour. Furthermore, we observed a similar behaviour when introducing a 'non-Hermitian defect' by selectively photoexciting the domain wall of the VPC, as shown in Fig. R5a. As the conductivity of Si increases due to photoexcitation, free carrier densities increase and imparts losses, consequently, the photoactive region acts as a 'non-Hermitian defect'.

Fig. R1 Transport of slow light in VPC waveguide through defects (a-c) The locus of Poynting vectors of slow light modes when the defects are introduced by varying the size of the triangular holes from $l_1 = 0.4a/\sqrt{3}$ to $l_1 = 0.55a/\sqrt{3}$, where $a = 260 \mu\text{m}$ is the periodicity of unit cell. **(d-f)** The locus of Poynting vectors of slow light modes when the defects are introduced by removing the triangular holes.

With regard to the references (Nature Photonics 17,386 (2023) and E. Verhagen, proceedings vol/ PC12196, Active Photonic Platforms, SPIE 2022) cited by reviewer, we recognize that these studies have examined the matter of topological protection of slow light modes in VPC waveguide. However, it is essential to highlight the key distinctions between the cited works and our own study, as outlined below:

- **Operating wavelength:** - The above cited works operate in the near-infrared region, where electromagnetic wave propagation is highly susceptible to nanometre-scale roughness or disorders. In contrast, our study focuses on terahertz waves ($\lambda \sim 1000 \mu\text{m}$), which exhibit relatively low sensitivity to roughness or defects on the scale of a few microns.
- **Propagation loss extraction method:** - The above-cited works employ the equation $\langle \ln T_{prop,L}(\lambda) \rangle = -\alpha(\lambda)L$, which assumes that the expectation value of log-transmittance through the VPC waveguide linearly depends on its length (L). Our study, however, uncovers that in the bearded interface VPC waveguide, the Poynting vector of kink states undergoes a detour mediated by the magnetic phase vortices. Consequently, for extracting the propagation loss in the VPC waveguide under slow light conditions, there is likely a need to modify the analytical expression of the transmission model. As such, we believe this could be one of the reasons that in the above-cited works the authors have not found any advantages of topological mode over the trivial mode in the slow light regime.

To clarify, our current work *unravels the mechanism of the slow light effect in the bearded interface VPC waveguide*. Nevertheless, to gain more comprehensive understanding, additional investigations are warranted, which we intend to explore in our subsequent research endeavours.

In consideration of reviewer's comments, we have introduced the discussion on the robustness of slow light modes in the revised manuscript and modified the figure, Fig 4 in the revised manuscript.

(From page 12, line 19 to page 13, line 25)

"We record the GD defined as $GD = -\frac{d\phi}{d\omega}$, (where ϕ and ω are phase and angular frequency of the transmitted wave, respectively) from the VPC waveguide chip using a vector network analyser (VNA) based experimental setup (see SI, section S5). We photoexcite the domain wall of VPC waveguide chip (see Fig. 3a) with a continuous wave laser of wavelength 532 nm to tune the slowness. The optical beam spot (diameter ≈ 0.5 mm) (see SI, Fig. S6 and S7) was kept small to ensure uniform illumination of few unit cells. Fig. 3b shows the variation of GD as a function of optical pump power. To accurately replicate the experimental conditions, we performed 3D simulation in CST Microwave Studio, as thoroughly explained in SI, section S7. Given that the optical beam spot in the experiment is focused on only a few unit cells (Fig. 3a), we emulate the effects of optical pumping by defining a circular region spanning a few unit cells (highlighted in red in Fig. S8) at the centre of the bearded interface of the VPC waveguide. Within this region, the conductivity of silicon is varied, which is equivalent to changing the optical pump power in experiment. Fig. 3b shows a comparison between the experimental and simulated GD curves, which shows good agreement in the variation of GD. Importantly, we observe an identical relative change (i.e., difference between maximum and minimum GD) in GD values for both the experiment and simulation, which is around 11%. The slight variations in GD values can be attributed to slight experimental misalignment and variations in the thickness (± 25 μm) of the fabricated VPC waveguide.

To understand the behaviour of GD as a function of optical pump power (or Si conductivity in simulation), we computed Poynting vector plots from 3D simulations. Fig. 3c-3e show the locus of Poynting vector calculated at the conductivity values marked in the inset of Fig. 3b, where the black circle encloses the region with variable Si conductivity. For $\sigma = 0$ S/m, the transport of kink state is accompanied by forward and counter-propagating waves, as depicted in Fig. 3c. As the conductivity of Si increases, free carrier densities increase and imparts losses, as a consequent the photoactive region acts as a non-Hermitian defect that attenuates the forward propagating waves, resulting in a decrease in GD, as shown in Fig. 3d. With further increase in the conductivity of Si, the kink state takes a detour around the circular region and retain its forward and counter-propagating waves after traversing the circular region (Fig. 3e), resulting an increase in GD, as evident in Fig. 3b. To ensure the consistency of our results, we conducted photoexcitation measurements multiple times by gradually increasing and decreasing the optical pump power. The results are depicted in Fig. S15, showing no variation in the measured GD values. This observation rules out the possibility of any hysteresis-type behaviour and validate the reproducibility in our experiment."

R1.2a The authors describe the physical mechanism behind slow light in the explored waveguides as that of coupling between closely spaced (adjacent) counter-propagating modes, which originate from magnetic phase vortices that have their locus at the small triangle in the VH photonic crystal unit cells. As much as I believe the physical picture is correct, the authors do not provide a sufficiently clear presentation of that physical picture:

- For example, the way in which the magnetic phase/Poynting vector vortices of single unit cells add up to lead to only forward propagating components in a zigzag interface and to both forward and backward propagating components in the bearded interface (Supplementary Sections S2 and S3) could be made more explicit. The reader is likely capable of constructing such a picture from the unit cell Poynting vector, but, and especially given that it is already in the Supplementary Material, it may help to have a succession of images where the progressive effect of adding vortices is observed.

Our response:

We appreciate the reviewer's suggestion and support for our proposed underlying physical picture of slow light in Valley Photonic Crystal (VPC) by saying "*I believe the physical picture is correct*". Regarding the reviewer's comment on deriving the forward and counter-propagating waves picture, we want to emphasize that the possibility of constructing the topological kink state Poynting vector from the unit cells is unlikely since the Poynting vectors of the unit cells can only be computed for frequencies of the bulk modes while the topological kink states span the frequencies within the band gap for which wave propagation are forbidden for the unit cells (or bulk VPC).

However, to enhance the clarity of our illustrations, we have improved the quality of Supplementary Videos (SV). In SVa and SVb, we show the Poynting vector flow at zigzag and bearded interface VPC waveguides, respectively. Additionally, we have included a new Supplementary Video, SVc, to demonstrate the evolution of the Poynting vector profile from "only forward propagating waves" to "forward and backward propagating waves" in the bearded interface waveguide by sweeping across the frequencies for which the group velocity of the topological kink state changes.

In consideration of reviewer's concern, we have modified the introduction part of the revised manuscript accordingly:

(On page 4, Line 25)

"For instance, zigzag interface with mirror symmetry yields only forward propagating kink states, whereas bearded interfaces with glide symmetry (reflection + translation) permit the coexistence of both forward and counter-propagating waves (see Supplementary Videos SVa and SVb). Noteworthy, for the appearance of counter-propagating waves in VPC waveguide, two conditions need to be satisfied: (a) the existence of magnetic phase vortices, and (b) the stacking of unit cells with a bearded interface with glide symmetry.

To demonstrate the significance of both these conditions for the existence of counter-propagating waves in the VPC, we have included a Supplementary Video SVc. This video demonstrates the evolution of the Poynting vector profile, transitioning from "only forward propagating waves" to "forward and counter-propagating waves" in the bearded interface VPC waveguide as we sweep across the corresponding frequencies from the K-valley to the band edge. As the magnetic phase vortices exist for wavevectors ranging from the projected K point ($k_x = 2\pi/3a$) to the band edge ($k_x = \pi/a$), we observe the appearance of counter-propagating waves from the frequency corresponding to the projected K point to the band edge frequency, as clearly evident in SVc."

R1.2b More importantly, the wording "counter-propagating modes" is used across all the manuscript, including its title, and the slow light effect is understood as coupling of these modes. Nevertheless, these two (or more depending on the transverse decay) modes are never really shown to exist independently. What I mean by this is that the authors provide no simulations of how these "modes" look like when only the involved magnetic phase vortices are included. I am unaware if this is possible, but I would encourage the authors to try something. The authors could maybe take regions that are forward-propagating or backward-propagating in the final waveguide, cut out that part in some clever way and embed it in a bulk of the same material. Then simulate the different waveguide "modes" and study the resulting waveguide dispersion as the (strong) coupling of these original modes. I understand that this might not work very well since the few-mode decomposition might be a very bad approximation. In any case, if such an approach is not possible, then I think calling these "counter-propagating modes" is misleading and maybe a better wording would be "counter-propagating adjacent wavefronts" or something along those lines.

Our response:

We thank the reviewer for providing very useful insights regarding the "counter-propagating modes" in bearded interface VPC waveguide. We recognize that the term "counter-propagating modes" might be misleading, as it could imply that the existence of two independent modes and interaction between them leads to the slow-light feature. Instead, we want to emphasize that the appearance of counter-propagating Poynting vector arises from two factors: a) the influence of magnetic phase vortices and b) the stacking of unit cell with a bearded interface, which preserve the glide symmetry (reflection + translation).

Following the reviewer's comment, we have replaced the term "counter-propagating modes" with "counter-propagating waves" throughout the manuscript to accurately describe the underlying physical phenomenon.

R1.3: With the physical picture at hand, the authors describe the effect of geometry and band gap size on the achieved slow light. Using that picture and the transverse decay of the topological modes perpendicular to the interface (which becomes faster as the gap becomes larger and therefore may control the coupling between the counter-propagating "modes"), they seek to explain the observed group velocities. Even if I understand that the C6 symmetry needs to be preserved, which limits the number of explored cases with the approach taken, I think the general conclusions drawn (though probably right) are not justified. Maybe a study for a fixed value of v and varying hole size asymmetry to control the gap would allow a more continuous sweep of the geometry and more well-supported conclusions. Let me elaborate on this: the authors claim that "examining $v_g^{(\pi/a)}$ reveals minimal change for Type I interface with respect to the bandgap, while contrasting alteration is observed for Type II interface, as clearly evident in the right inset of Fig. 2a-2c." While I agree that such value seems more stable for Type I, I find that the conclusion drawn from it, which is that slow light in Type I waveguides is relatively unimpacted by the band gap width because the counter-propagating regions are in close proximity and are therefore less affected by the transverse decay, is done based on only a few band structures and on a single wavevector. I fail to understand in which way the Brillouin zone edge is a representative reciprocal point in this work, and I would rather expect that, if the physical picture holds, it does also relatively well at other reciprocal vectors. Actually, the only wavevector that may carry a particular meaning for these structures is the projection of the K point along the interface direction, somewhere around $0.7\pi/a$, or the region in its vicinity, where topological properties are guaranteed.

Our response:

We thank the reviewer for providing the detailed comments. Following the reviewer's suggestion, we varied the asymmetry of the unit cell with $v = 6$, to continuously tune the bandgap of the VPC. Fig. R2a shows the bandgap of VPCs as a function of air hole radii (r_1, r_2) for hexagonal air holes ($v = 6$) of period $a = 260 \mu\text{m}$. The colormap represents the strength of bandgap, from which we selected the unit cells with increasing bandgap, as marked in Fig. R2a. The specific radii (r_1, r_2) of selected unit cells are as follows: $(0, 0.79a/\sqrt{3})$, $(0.22a/\sqrt{3}, 0.78a/\sqrt{3})$, $(0.29a/\sqrt{3}, 0.75a/\sqrt{3})$, $(0.33a/\sqrt{3}, 0.71a/\sqrt{3})$ and $(0.37a/\sqrt{3}, 0.67a/\sqrt{3})$, with corresponding bandgap (Δf) 166 GHz, 141 GHz, 115 GHz, 84 GHz and 61 GHz, respectively.

Increasing the bandgap of VPC leads to increased transverse decay of the topological modes perpendicular to the interface, resulting in the enhancement of group velocity (v_g) of the kink states due to weak coupling between forward and counter-propagating waves. To further elaborate, we have plotted the group velocity extracted at Brillouin zone edge ($v_g^{\pi/a}$ where $k_x = \pi/a$) for Type I and Type II bearded interfaces in Fig. R2b. In the case of Type I bearded interface, the proximity of the adjacent counter-propagating waves to the interface causes the slowness (or $v_g^{\pi/a}$) to remain relatively unchanged, even with an increase in bandgap, as shown by red solid line in Fig. R2b. Conversely, for Type II bearded interface, increasing the bandgap reduces the spatial extent of the topological modes perpendicular to the interface, leading to weak coupling between the forward and adjacent counter-propagating waves. Consequently, $v_g^{\pi/a}$ increases with the increase in bandgap as shown by blue solid line in Fig. R2b. This is further corroborated through the Poynting vector (\vec{S}) plots depicted in Fig. R2d-2h, where we observe gradual decrease in the strength of the adjacent counter-propagating waves with an increase in bandgap (from left to right), which results in an increase in $v_g^{\pi/a}$.

Moreover, to demonstrate the universality of the underlying mechanism of slow light in bearded interface VPC waveguide, which arises from the interaction between forward and adjacent counter-propagating waves, we calculated the group velocity at the projected K valley i.e., ($v_g^{2\pi/3a}$ where $k_x = 2\pi/3a$), as

shown in Fig. R2c. Similar to the previous case, there is a lack of correlation between $v_g^{2\pi/3a}$ and the bandgap for Type I bearded interface while $v_g^{2\pi/3a}$ increases with the bandgap for Type II interface. This observation is further supported by the Poynting vector plot shown in Fig. R2i-2m, where increasing the bandgap reduces the strength of adjacent counter-propagating waves, leading to an increase in group velocity.

Fig. R2 Tuning the slowness of VPC via bandgap engineering (a) The bandgap of VPCs as a function of air hole radii (r_1, r_2) for hexagonal air holes ($v = 6$) of period $a = 260 \mu m$. The colormap indicates the strength of bandgap, while markers represent the selected unit cells with increasing bandgap. The specific radii (r_1, r_2) of the chosen unit cells are as follows: $(0, 0.79a/\sqrt{3})$, $(0.22a/\sqrt{3}, 0.78a/\sqrt{3})$, $(0.29a/\sqrt{3}, 0.75a/\sqrt{3})$, $(0.33a/\sqrt{3}, 0.71a/\sqrt{3})$ and $(0.37a/\sqrt{3}, 0.67a/\sqrt{3})$, with corresponding bandgap (Δf) values 166 GHz, 141 GHz, 115 GHz, 84 GHz and 61 GHz, respectively. (b) Band edge group velocity ($v_g^{\pi/a}$) is calculated for Type I (red solid line) and Type II (blue solid line) bearded interface VPC waveguides, shown on the left Y-axis. The bandgap corresponding to the selected unit cells is shown on the right Y-axis. (c) Group velocity is extracted at projected K-value (i.e., $k_x = 2\pi/3a$) for Type I and Type II bearded interface VPC waveguides, shown on the left Y-axis. (d-h) The locus of the Poynting vector is extracted at $k_x = \pi/a$. The bandgap increases from 61 GHz to 166 GHz from left to right. (i-m) The locus of the Poynting vector is extracted at projected K-valley i.e., $k_x = 2\pi/3a$. The bandgap increases from 61 GHz to 166 GHz from left to right.

We have included these updated results in the revised manuscript as **new Fig. 3** and modified the discussion section as follow:

(From page 9, line 17 to page 10, line 21)

“To further emphasize the effect of the band gap on v_g , we conducted an extensive investigation by fixing the air hole shape to be a hexagon ($\nu = 6$) and varied the air hole radii (r_1, r_2) of period $a = 260 \mu\text{m}$. Fig. 3a presents the band gap of VPCs, with the colormap representing the strength of the band gap. We selected the unit cells with increasing band gap, as marked in Fig. 3a. The specific radii (r_1, r_2) of these chosen unit cells are as follows: $(0, 0.79a/\sqrt{3})$, $(0.22a/\sqrt{3}, 0.78a/\sqrt{3})$, $(0.29a/\sqrt{3}, 0.75a/\sqrt{3})$, $(0.33a/\sqrt{3}, 0.71a/\sqrt{3})$ and $(0.37a/\sqrt{3}, 0.67a/\sqrt{3})$, with corresponding band gap (Δf) 166 GHz, 141 GHz, 115 GHz, 84 GHz and 61 GHz, respectively. The projected band diagrams for these chosen unit cells are depicted in Fig. 3d-3h, showing a clear increase in the band gap from left to right. Fig. 3b shows the group velocity extracted at Brillouin zone edge ($v_g^{\pi/a}$ where $k_x = \pi/a$) for Type I and Type II bearded interface VPC waveguides. For Type I bearded interface, the proximity of the counter-propagating wavefront to the interface causes the slowness (or $v_g^{\pi/a}$) to remain relatively unchanged, even with an increase in band gap, as shown by red solid line in Fig. 3b. In contrast, for Type II bearded interface, increasing the band gap reduces the spatial extent of the topological modes perpendicular to the interface, leading to weak coupling between the forward and adjacent counter-propagating waves. Consequently, $v_g^{\pi/a}$ increases with the increase in band gap as shown by blue solid line in Fig. 3b. This observation is further supported by the Poynting vector (\vec{S}) plots depicted in Fig. 3i-3m, where we observe gradual decrease in the strength of the counter-propagating waves with an increase in band gap (from left to right) for Type II interface, which results in an increase in $v_g^{\pi/a}$.

Moreover, to demonstrate the universality of the underlying mechanism of slow light in bearded interface VPC waveguide, which arises from the interaction between forward and counter-propagating waves, we calculated the group velocity at the projected K valley i.e., ($v_g^{2\pi/3a}$ where $k_x = 2\pi/3a$), as shown in Fig. 3c. Similarly, there is no correlation between $v_g^{2\pi/3a}$ and the band gap for Type I bearded interface while $v_g^{2\pi/3a}$ increases with the band gap for Type II interface, which is also supported by the Poynting vector plots (Fig. S4).”

R1.4a: Trying to use also the same physical picture, the authors study how to control the group velocity by locally pumping (down to the size of the beam spot) some region of the waveguide and, as a consequence, controlling the coupling between counter-propagating "modes". Regarding this experiment and its comparison to simulations, I also have a number of remarks:

- The authors compare the overall evolution of the experimentally evaluated group index with pump power to the simulated values of the group index as a function of the conductivity of silicon. They then claim that "the simulated ng exhibits large value compared to experiment" and that "it is attributed due to periodic boundary condition used in the simulation, which uniformly changes the conductivity of Si throughout bearded interface waveguiding region, whereas in experiment the photoexcitation of Si is limited to just one- or two-unit cells." While I might understand that the boundary conditions are important, the disparity is not only huge but the value for no pump power and $\sigma = 0$ does not seem to match at all, which invalidates their argument. I think the local nature of the excitation (experiment) compared to the periodic nature of the excitation (simulation) is not the reason behind this, although it does certainly play a role. One aspect that is critical when dealing with the evaluation of the group index is which method is used. Are the authors taking a numerical derivative or are they using a formula for it (see for example Eq. 10 in [Lalanne et al. ACS Photonics 9, 4 (2019)], where the group velocity can be found at a specific wavevector using Lorentz reciprocity)? If they are using a numerical derivative, I suggest that they use the formula. The problem with evaluating the group index using a numerical derivative in glide-plane-symmetric waveguides is that, unless the authors have enforced the glide-plane symmetry to the finite-element mesh, the resulting band structure will have a small gap at the degeneracy point and therefore exhibit a diverging group index at the band edge, i.e., a very large group index when evaluated using the derivative. I have personally had this same issue, which I solved by either an ultra-fine mesh or via such formulas. Given that the evaluations are done at $k = \pi/a$, the very large values they report suspiciously indicate this issue. Somehow, the fact that the PWE simulation with a complex refractive index (roughly the same as having a finite conductivity...) gives much better results also indicates the previous. If the authors have, on the contrary, not used a numerical derivative

to calculate n_g , then I would encourage them to revisit their COMSOL band structure model and/or give more details than those in Supplementary Section S10.

Our response:

We thank the reviewer for providing detailed feedback. To address the discrepancy between the experimental results involving optical pumping and simulation, we opted to compare the group delay (GD) instead of the group index (n_g), for the following reasons:

- GD directly quantifies the time delay experienced by signal or electromagnetic waves as they propagate through the waveguide. On the other hand, group index (n_g) is a measure of the refractive index of a waveguide, considering the dispersion of the waveguide.
- In the case of photoexcitation, the optical pump spot spans up to a few unit cells, inducing free carrier locally, which in turn effectively leads to changes in the refractive index of these few unit cells. As the group index (n_g) is a global parameter that encapsulates the effective refractive index of the entire waveguide, n_g is not a good metric to characterize the localized changes induced by photoexcitation.

To emulate the experimental conditions, we performed a full 3D simulation in CST Microwave Studio. The simulation setup, depicted in Fig. R3a, includes adiabatic couplers designed at both the input and output facets of the VPC waveguide. To couple the terahertz wave into the VPC waveguide, we used a rectangular port (port 1) connected to the WR 2.8 metallic waveguide. The output signal was collected from the output port (port 2 in Fig. R3a) using an identical WR 2.8 metallic waveguide, as depicted in Fig. R3a.

To emulate the effects of optical pumping, we defined a circular region (highlighted in red in Fig. R3a) within VPC waveguide, spanning a few unit cells of bearded interface with variable silicon conductivity. To ensure high accuracy in simulation, we implemented sufficiently dense meshing, particularly in the region where silicon conductivity varies, as shown in Fig. R3b and R3c.

Fig. R3 3D simulation setup in CST Microwave Studio (a) Schematic of Type I bearded VPC waveguide with $\nu = 3$ in CST simulation environment. Adiabatic couplers are utilized to efficiently couple the THz waves in and out of the VPC waveguide. Rectangular ports (port 1 and port 2) are connected to WR 2.8 waveguides to launch and receive the THz signals. The red region in the diagram represents silicon with variable conductivity (b and c) The mesh views show a close-up of the bearded interface and the spot where the optical pump (i.e., region with variable silicon conductivity) is located, respectively. The entire simulation is carried out using a *time domain solver*.

Fig. R4a shows the simulated transmission and GD spectra which exhibits close resemblance to the experimentally recorded spectra (Fig. R4b). However, due to slight experimental misalignment and variations in the thickness ($\pm 25 \mu\text{m}$) of the fabricated VPC waveguide, there are minor differences observed in the transmission amplitude and GD (marked by a star symbol).

Fig. R4 Transmission and group delay (GD) spectra (a and b) Simulated and experimental transmission (red solid line) and GD (blue solid line) spectra. The star mark represents band edge frequency for which GD is extracted.

Following the reviewer's suggestion, we have included the updated results in the new **Fig. 4** and updated the text in the revised manuscript as:

(From page 12, line 13 to page 13, line 25)

"To experimentally quantify the slowness of kink states, we opted to compare the group delay (GD) instead of the group index (n_g). This is because GD directly quantifies the time delay experienced by signal or electromagnetic waves as they propagate through the waveguide. While n_g is a global parameter that encapsulates the effective refractive index of the entire waveguide and does not accurately reflect the localized changes in the refractive index induced by optical pumping.

We record the GD defined as $GD = -\frac{d\phi}{d\omega}$, (where ϕ and ω are phase and angular frequency of the transmitted wave, respectively) from the VPC waveguide chip using a vector network analyser (VNA) based experimental setup (see SI, section S5). We photoexcite the domain wall of VPC waveguide chip (see Fig. 3a) with a continuous wave laser of wavelength 532 nm to tune the slowness. The optical beam spot (diameter ≈ 0.5 mm) (see SI, Fig. S6 and S7) was kept small to ensure uniform illumination of few unit cells. Fig. 3b shows the variation of GD as a function of optical pump power. To accurately replicate the experimental conditions, we performed 3D simulation in CST Microwave Studio, as thoroughly explained in SI, section S7. Given that the optical beam spot in the experiment is focused on only a few unit cells (Fig. 3a), we emulate the effects of optical pumping by defining a circular region spanning a few unit cells (highlighted in red in Fig. S8) at the centre of the bearded interface of the VPC waveguide. Within this region, the conductivity of silicon is varied, which is equivalent to changing the optical pump power in experiment. Fig. 3b shows a comparison between the experimental and simulated GD curves, which shows good agreement in the variation of GD. Importantly, we observe an identical relative change (i.e., difference between maximum and minimum GD) in GD values for both the experiment and simulation, which is around 11%. The slight variations in GD values can be attributed to slight experimental misalignment and variations in the thickness (± 25 μm) of the fabricated VPC waveguide.

To understand the behaviour of GD as a function of optical pump power (or Si conductivity in simulation), we computed Poynting vector plots from 3D simulations. Fig. 3c-3e show the locus of Poynting vector calculated at the conductivity values marked in the inset of Fig. 3b, where the black circle encloses the region with variable Si conductivity. For $\sigma = 0$ S/m, the transport of kink state is accompanied by forward and counter-propagating waves, as depicted in Fig. 3c. As the conductivity of Si increases, free carrier densities increase and imparts losses, as a consequent the photoactive region acts as a non-Hermitian defect that attenuates the forward propagating waves, resulting in a decrease in GD, as shown in Fig.3d. With further increase in the conductivity of Si, the kink state takes a detour around the circular region and retain its forward and counter-propagating waves after traversing the circular region (Fig. 3e), resulting an increase in GD, as evident in Fig. 3b. To ensure the consistency of our results, we conducted photoexcitation measurements multiple times by gradually increasing and decreasing the optical pump power. The results are depicted in Fig. S15, showing no variation in the measured GD values. This observation rules out the possibility of any hysteresis-type behaviour and validate the reproducibility in our experiment."

In addition, the extended discussion on simulation is presented in the supplementary information, **section S7 and S9.**

R 1.4b: If we disregard the disparity of the values of the group index and focus only on the shape of its evolution with excitation, the authors claim that the simulation behaviour is "akin to experiment". However, I feel that such a statement is not true. The only aspect where simulation and experiment match is that the group index increases initially. However, the experiment seems to exhibit a linear

growth of n_g , but not the simulation. After that initial growth, in one case there is an extremely fast (step-like) drop and in the other a quite smooth evolution going across a maximum. In addition, the experiment shows that the group index grows again after that drop. I think a wider experimental data set (measurements on many more waveguides), a reproducibility study and an analysis of possible hysteresis (is there any permanent damage?) would clarify what that drop is. The authors attribute it to "the increased photoexcitation at the magnetic phase vortex singularity, destroying the supported adjacent counter-propagating modes". This might be indeed true, but it is only a hypothesis and, given that the actuation mechanism is supposed to build upon the counter-propagating "modes" model, a hypothesis might not be enough. Finding better proof of this would contribute a lot to the coherence of this part of the manuscript. One possibility would be for the authors to elaborate a slightly more advanced band structure model where instead of a single slice with a finite conductivity/extinction coefficient, they take several slices with changes in extinction coefficient that follow a Gaussian excitation spot (for PWE) or a spatially dependent conductivity following a gaussian excitation spot (for COMSOL). In that case, the pump power increase will be better mimicked through simulation.

Our response:

In the response to the reviewer's comment and to reconcile the differences between the experiment and simulation, we opted to utilize the group delay (GD) rather than the group index (n_g). This is because group index represents a global parameter and does not accurately reflect the localized changes in the refractive index induced by optical pumping.

To accurately replicate the experimental conditions, we performed 3D simulation in CST Microwave Studio, as thoroughly explained in the response to the previous comment (R1.4a). Fig. R5a shows the optical image of the fabricated VPC waveguide of Type I bearded interface. The waveguiding path is highlighted by blue shaded region. To illustrate the achieved active tunability, Fig. R5b shows a comparison between the experimental and simulated GD curves, which shows good agreement in the variation of GD. Given that the optical beam spot in the experiment is focused on only a few unit cells, we emulate the effects of optical pumping by defining a circular region spanning a few unit cells (highlighted in black in Fig. R5c-R5e) at the centre of the bearded interface of the VPC waveguide. Within this region, the conductivity of silicon can be varied, which is equivalent to changing the optical pump power in experiment. Importantly, we observe an identical relative change (i.e., difference between maximum and minimum GD) in GD values for both the experiment and simulation, which is around 11%. The differences in the experimental and simulated GD values can be attributed to slight experimental misalignment and variations in the thickness ($\pm 25 \mu m$) of the fabricated VPC waveguide.

The variation of GD as a function of optical pump power (or silicon conductivity in simulation) can be understood from the Poynting vector plots. Fig. R5c-R5e show the locus of Poynting vector calculated from 3D simulation at the conductivity values marked in the inset of Fig. R5b, where the black circle encloses the region with variable silicon conductivity. For $\sigma = 0 S/m$, the transport of kink state is accompanied by forward and counter-propagating waves, as depicted in Fig. R5c. As the conductivity of Si increases, free carrier densities increase and imparts losses, consequently the photoactive region acts as a 'non-Hermitian defect' which attenuates the counter-propagating waves, resulting in a decrease in group delay, as shown in Fig. R5d. With a further increase in the conductivity of silicon, the kink state takes a detour around the circular region and retain its forward and counter-propagating moving waves after traversing the circular region (Fig. R5e), resulting an increase in group delay, as evident in Fig. R5b.

Fig R5 Active tuning of group delay (GD) (a) Optical image of fabricated Type I bearded interface VPC waveguide with $\nu = 3$ air holes. All critical dimensions are highlighted. The green circular region indicates the location of the optical pump spot ($\lambda = 532 \text{ nm}$). (b) The experimental GD is plotted as a function of optical pump power. The GD values are obtained using a vector network analyzer (VNA) based setup. The inset graph illustrates the simulated GD against the conductivity of silicon, which shows good agreement with experimental results. For the simulation, a 3D approach is employed in CST Microwave Studio to replicate the experimental conditions. (c-e) The locus of Poynting vectors is calculated at different conductivity of silicon, as marked in the inset of graph 'b'. The black circle encloses the region with variable silicon conductivity.

Furthermore, to ensure the consistency of our results, we conducted photoexcitation measurements multiple times on different fabricated samples by gradually increasing and decreasing the optical pump power. The results are depicted in Fig. R6, showed no variation in the measured GD values. This observation rules out the possibility of any hysteresis-type behaviour in our experiment.

Fig. R6 GD as a function of optical pump power

Following the reviewer’s suggestion, we have included the updated results in the new **Fig. 4** and SI, section S10 and updated the text the revised manuscript as:

(From page 12, line 19 to page 13, line 25)

“We record the GD defined as $GD = -\frac{d\phi}{d\omega}$, (where ϕ and ω are phase and angular frequency of the transmitted wave, respectively) from the VPC waveguide chip using a vector network analyser (VNA) based experimental setup (see SI, section S5). We photoexcite the domain wall of VPC waveguide chip (see Fig. 3a) with a continuous wave laser of wavelength 532 nm to tune the slowness. The optical beam spot (diameter ≈ 0.5 mm) (see SI, Fig. S6 and S7) was kept small to ensure uniform illumination of few unit cells. Fig. 3b shows the variation of GD as a function of optical pump power. To accurately replicate the experimental conditions, we performed 3D simulation in CST Microwave Studio, as thoroughly explained in SI, section S7. Given that the optical beam spot in the experiment is focused on only a few unit cells (Fig. 3a), we emulate the effects of optical pumping by defining a circular region spanning a few unit cells (highlighted in red in Fig. S8) at the centre of the bearded interface of the VPC waveguide. Within this region, the conductivity of silicon is varied, which is equivalent to changing the optical pump power in experiment. Fig. 3b shows a comparison between the experimental and simulated GD curves, which shows good agreement in the variation of GD. Importantly, we observe an identical relative change (i.e., difference between maximum and minimum GD) in GD values for both the experiment and simulation, which is around 11%. The slight variations in GD values can be attributed to slight experimental misalignment and variations in the thickness (± 25 μm) of the fabricated VPC waveguide.

To understand the behaviour of GD as a function of optical pump power (or Si conductivity in simulation), we computed Poynting vector plots from 3D simulations. Fig. 3c-3e show the locus of Poynting vector calculated at the conductivity values marked in the inset of Fig. 3b, where the black circle encloses the region with variable Si conductivity. For $\sigma = 0$ S/m, the transport of kink state is accompanied by forward and counter-propagating waves, as depicted in Fig. 3c. As the conductivity of Si increases, free carrier densities increase and imparts losses, as a consequent the photoactive region acts as a non-Hermitian defect that attenuates the forward propagating waves, resulting in a decrease in GD, as shown in Fig. 3d. With further increase in the conductivity of Si, the kink state takes a detour around the circular region and retain its forward and counter-propagating waves after traversing the circular region (Fig. 3e), resulting an increase in GD, as evident in Fig. 3b. To ensure the consistency of our results, we

conducted photoexcitation measurements multiple times by gradually increasing and decreasing the optical pump power. The results are depicted in Fig. S15, showing no variation in the measured GD values. This observation rules out the possibility of any hysteresis-type behaviour and validate the reproducibility in our experiment.”

In addition, the extended discussion on simulation is presented in the supplementary information, section S7 and S9.

R 1.4c: Section S9 seems quite interesting although I fail to understand exactly how the simulation is set. Are the authors using an input and output port in COMSOL or is it also a periodic simulation with the laser spot effect at the centre of the supercell? Please, provide more details than what is in the Methods section. Whatever the details, I find this much more relevant than all the band structure simulations the authors have done, since these simulations are much closer to the experiment they perform. What is really occurring in the experiment is not a group index change or “an active control of slow light”, but the creation of a lattice-scale defect through which the topological kink state can travel through. Since this would be the actual underlying mechanism, I would suggest this is brought forward to the main text and compared to what happens for the trivial branch. That comparison, completely lacking in the manuscript, is very relevant, especially if they want to highlight the benefits of using a topological mode in view of recent negative results on their use to fight backscattering from fabrication imperfection.

Our response:

We very much appreciate the reviewer’s suggestion and thank him/her for applauding our results shown in section S9. In our study, we conducted a 2D simulation using COMSOL Multiphysics to analyze the Poynting vector flow in the VPC waveguide by locally varying the conductivity of silicon near to the bearded interface, as shown in Fig. R7a. Similar to 3D simulation (Fig. R5a), the terahertz waves are coupled into the VPC waveguide using an input port connected to the WR 2.8 waveguide with an adiabatic taper coupler inserted, as depicted in Fig. R7a.

To replicate the behaviour of the WR 2.8 metallic waveguide in COMSOL, we employed perfect electric conductor (PEC) boundary condition at the edges (Fig. R7a). For the simulation process, we utilized the electromagnetic waves, frequency domain (ewfd) module of COMSOL Multiphysics. To ensure accuracy, we implemented user assigned mesh setting, and define the maximum and minimum mesh element sizes as 25 μm and 5 μm , respectively.

Fig R7: 2D simulation setup in COMSOL Multiphysics (a) Schematic of 2D simulation setup in COMSOL Multiphysics. Perfect electric conductor (PEC) is employed at the edges of rectangle to mimic the WR 2.8 metallic waveguide. Numeric ports are defined at both the input and out facets of the WR 2.8 waveguide. The VPC waveguide is surrounded by a scattering boundary condition to eliminate the effect of scattered waves. The black circle represents the area in VPC waveguide where silicon conductivity is variable. (b and c) The mesh views of air holes and air holes with silicon area, respectively. To ensure accuracy, we implemented user assigned mesh setting, and define the maximum and minimum mesh element sizes as $25\ \mu\text{m}$ and $5\ \mu\text{m}$, respectively.

Fig. R8 shows the simulated transmission spectra from COMSOL Multiphysics. A bearded interface supports both topological and trivial mode due to the presence of glide symmetry. Building upon the ability of valley kink states to overcome sharp bends, we eliminate the contribution of trivial states in the transmission spectrum by constructing Z-shaped domain walls featuring two sharp bends. As a result, the transmission corresponding to the trivial mode is effectively suppressed in the waveguide with bends, leading to a reduced spectral bandwidth, as depicted by red shaded region in Fig. R8.

Fig. R8 Simulated transmission spectra from the VPC waveguide for both the straight (black solid line) and bend (red solid line) bearded interface configurations. In the bend waveguide, the transmission associated with the trivial modes is suppressed. The red and black shades represent the topological and trivial regions, respectively.

Additionally, to explore the impact of photoexcitation on the transport properties of both topological and trivial modes, we conducted Poynting vector simulations using COMSOL. Within this analysis, we introduced a circular region spanning the bearded interface with varying silicon conductivity. The Poynting vectors were simulated at two distinct frequencies corresponding to topological and trivial modes, as highlighted by the vertical dashed line in Fig. R9.

Fig. R9 shows the simulated Poynting vector for different silicon conductivity values for both topological and trivial modes. The black circle encloses the region with variable silicon conductivity. Upon investigating the evolution of Poynting vectors for the topological mode (Fig. R9a-R9c), it becomes apparent that the kink state takes a detour around the circular region and retain its forward and counter-propagating waves after traversing the circular region. In contrast, for the trivial mode, there is a significant scattering and back reflection with an increase in silicon conductivity, as evident in Fig. R9d-9f.

Fig R9: Locus of Poynting vector simulated in COMSOL Multiphysics (a-c) The Poynting vector is simulated for the frequency corresponding to the topological mode (i.e., 0.325 THz), marked by red dashed vertical line in Fig. R8. The region of the VPC with variable silicon conductivity is enclosed within a black circle. From left to right, the conductivity of silicon increases from 0 S/m to 50 S/m. **(d-f)** The Poynting vector is simulated for the frequency corresponding to the trivial mode (i.e., 0.34 THz), marked by the black dashed vertical line in Fig. R8. Similar to previous case, the region of VPC with variable silicon conductivity is enclosed within a black circle. The conductivity of silicon increases from 0 S/m to 50 S/m from left to right.

In consideration to the reviewer's comment, we have updated the supplementary information, section S9.

In addition to the more fundamental issues just discussed, a number of small technical points, presentation issues or less-important remarks should also be addressed:

R1.5: In L40, the wording "extreme sensitivity to disorders accompanied by fabrications" reads strange. Maybe the authors could rephrase it to "extreme sensitivity to disorder resulting from the fabrication process".

We appreciate the reviewer's suggestion. As pointed out we have rephrased the highlighted sentence in the revised manuscript. Now the rephrased text is reflected at page 2, line 8 in the revised manuscript.

R1.6: The rendering of Figs. 1 and 2 is very poor (in general of all figures) and it becomes hard to see some of the features, e.g., the Poynting vector field lines in 1(a). The review process will benefit a lot from getting accurately rendered images, otherwise, a large portion of the information conveyed by figures is just lost.

We thank the reviewer for pointing this out. To solve the rendering problem, we have improved the quality of the images in the modified version of manuscript.

R1.7: L86: "the" photonic bandgap reads strange, maybe "a" photonic bandgap is more accurate.

We thank the reviewer for the suggestion. In the revised manuscript we have fixed all the minor grammatical errors.

R1.8: The PWE simulations are 2D but the structure is inherently 3D, how good is the agreement? In principle, the PWE could be extended to 3D, right?

We thank the reviewer for the question. Several previous studies have shown that the results of 2D simulation accurately matches the 3D case by employing a 2D-to-1D effective index approximations of photonic crystal slabs¹ (*Hammer, M. & Ivanova, O. V. Effective index approximations of photonic crystal slabs: a 2-to-1-D assessment. Opt Quant Electron 41, 267–283 (2009).*). For a slab thickness of **200 microns**, a refractive index of 3.42 for silicon can be approximated as 3.0 in the 2D case².

However, please note that we have opted to compare GD instead of n_g . As a result, all the 2D PWE simulations are replaced by 3D CST simulations in the new **Fig. 4**.

R1.9: In L167-173, the authors make the distinction between the two modes as topological and trivial. Can they comment on how this distinction is made? One of the geometries ($\nu = 3$ and Type I interface) is explored experimentally and the use of 120° bends may be used to identify the topological mode in that one, but the authors should describe their generic method to identify which mode is topological or refer to the appropriate reference if existent.

Our response:

We thank the reviewer for highlighting this out. We concur with the reviewer’s insight that, for $\nu = 3$ and Type I interface, the introduction of sharp bends could serve as a mean to discern the topological mode, as exemplified in Fig. R8. However, to accurately identify the topological and trivial modes within the bearded interface VPC waveguide, it is appropriate to observe the counter-propagating waves. For example, in case of $\nu = 3$, Type II bearded interface VPC waveguide, we identified the topological and trivial modes by examining the Poynting vectors at K- valley, as depicted in Fig. R10a and R10b. It is evident that for $\nu = 3$ and Type II bearded interface, the lower band is the topological band as there exists counter-propagating waves (Fig. R10a) whereas the upper band signifies the trivial band as evidenced by the absence of counter-propagating waves in the Poynting vector (Fig. R10b).

Fig. R10 Identification of topological and trivial mode in $\nu = 3$, Type II bearded VPC waveguide (a and b)
The locus of Poynting vectors corresponding to lower and upper band, respectively.

Similarly, we extended this procedure for $\nu = 9$ Type I and Type II bearded interface VPC waveguide as depicted in Fig. R11 and R12, respectively.

Fig. R11a and R11d show the projected band diagram for $\nu = 9$, Type I bearded interface, where the topological and trivial modes are indicated by solid and dashed line, respectively. Correspondingly, Fig. R11b and R11e show the transmission spectra through the $\nu = 9$, Type I VPC waveguide, where the combination of solid and dashed lines highlights the transmission corresponding to topological and trivial modes, respectively. Upon careful analysis of the Poynting vector linked to the topological modes, we observe counter-propagating waves (blue arrows) which arises due to magnetic phase vortices and signifies the topological features, as shown in Fig. R11c and R11f.

Fig. R11 Identification of topological and trivial mode in $\nu = 9$, Type I bearded VPC waveguide (a and d)
The projected band diagram (b and e) Simulated transmission spectra. The red dot marks the trivial and topological modes for which Poynting vectors are simulated in 'c' and 'f' (c and f) The locus of Poynting vectors corresponding to trivial and topological modes, respectively.

Fig. R12a and R12d show the projected band diagram for $\nu = 9$, Type II bearded interface VPC waveguide, where the topological and trivial modes are highlighted by solid and dashed lines, respectively. Fig. R12b and R12e depict the transmission spectra, where the hybridization of solid and dash lines highlights the contribution corresponding to topological and trivial modes, respectively. Likewise, examining the locus of Poynting vectors corresponding to topological mode reveal the presence of counter-propagating waves (Fig. R12c), while trivial mode does not exhibit such counter-propagating waves (Fig. R12f).

Fig. R12 Identification of topological and trivial mode in $\nu = 9$, Type II bearded VPC waveguide (a and d) The projected band diagram (b and e) Simulated transmission spectra. The blue dot marks the trivial and topological modes for which Poynting vectors are simulated in 'c' and 'f' (c and f) The locus of Poynting vectors corresponding to topological and trivial modes, respectively.

In the response to the reviewer's suggestion, we have taken the opportunity to include a description on the identification of topological and trivial modes. Furthermore, we have included illustrative videos SVd and SVe and have updated the supplementary information, section S1. The revised manuscript now incorporates the following updated text:

(On page 7, line 20)

"It is noteworthy that, to precisely distinguish between the topological and trivial modes within the bearded interface VPC waveguide, it is appropriate to observe the presence of counter-propagating waves (see detailed discussion in SI, section S1)."

R1.10: The Supplementary Video SVa will benefit from some more explanation, e.g., labelling in the video itself.

Our response:

Following the reviewer's suggestion, we have improved the quality of all Supplementary Videos (SVa, SVb, SVc, SVd and SVe) which includes proper labelling and slow frame speed for better visualization.

R1.11: There is probably a sign mistake in the group velocity plot of Fig. 2a, since the value at $k = \pi/a$ is in principle negative. This sign does not change the core message of the manuscript.

Our response:

We thank the reviewer for pointing this out. In the revised manuscript we have corrected the inconsistency in the sign of the group velocity of Fig. 2.

R1.12: In L227, some reference regarding transmission across a Z-shaped domain is needed.

Our response:

We thank the reviewer for pointing this. In the revised manuscript, we have cited the following references: *Nature Photonics* 14, 446–451 (2020), *Opt. Express*, OE 29, 13441–13450 (2021), *Opt. Lett.*, OL 45, 2648–2651 (2020) and *Nature Nanotechnology* 14, 31–34 (2019). It appears as citation, 21,22, 27 and 28 in the revised manuscript.

R1.13: - In Fig. S7, can the authors provide a wider span, where the transmission of the trivial band is shown, or they were limited by the VNA range?

Our response:

We fabricate the $\nu = 3$, Type I bearded VPC waveguide with Z-shape domain walls (see Fig. R5a) to eliminate the contribution from trivial bands. As a result, we observe sharp dip in transmission at 348 GHz. However, in the response of reviewer, we present the transmission curve for wide frequency range (320-360 GHz) in Fig. R13.

Fig. R13: Experimentally recorded transmission curve from $\nu = 3$, Type I bearded VPC waveguide with Z-shaped domain walls.

R1.14: It seems to me that a full analysis of the group index curve as a function of frequency could be done the same way that they do the analysis for the maximum ($k = \pi/a$). It could provide a much more complete picture of the effect of photoexcitation and whether the modelling can capture it correctly (the modelling is done as shown by the group index curves in Fig. 3d).

Our response:

Following the reviewer's suggestions in the previous comments (R1.4a), we have decided to compare the experimentally measured and simulated group delay (GD) instead of group index (n_g). We have replaced the PWE simulations with 3D CST simulations which allow us to directly compute the group delay. As per Fig. R5, our experimental and simulated group delay is in good agreement. Moreover, to enrich the discussion of active tuning of group index, we elaborated the PWE analysis in detail in the supplementary information, section S8.

Reviewer #2 (Remarks to the Author):

In their study, Kumar et al demonstrated slow light in a topological photonic system that resembles the quantum valley Hall effect in the terahertz regime. While there has been a previous demonstration of this concept by Yoshimi et al (Ref. 22 in the manuscript), the authors were able to elucidate the

underlying mechanism of the slow light, which is found to be the counter propagating energy flows induced by topological vortices distributed along the interface. Additionally, the authors experimentally demonstrated that the slow factor can be tuned by optical pumping. The overall quality of the work is good. I am happy to recommend its publication in Nature Communications if the following points can be addressed.

We sincerely thank the reviewer for positive feedback on our work, acknowledging that “*the overall quality of the work is good. I am happy to recommend its publication in Nature Communications*” and supporting it for publication. Below, we address the reviewer’s comments in a pointwise manner to further improve the manuscript.

R2.1: What does ‘kink’ state mean? The authors should provide some description in the text.

Our response:

We thank the reviewer for the comment. In valley photonic crystal (VPC) the domain wall, whether in zigzag or bearded configuration, is formed by stacking two mirror image sublattices (Type A and Type B), which introduces a sharp kink in the order parameter. In the field of ‘topological photonics’ the ‘topological edge state’ at domain wall of VPC is commonly referred as ‘kink state’, as it has been mentioned in several previously reported works³⁻⁵ as well.

In the response to the reviewer’s suggestion, we have taken the opportunity to include a description of the kink state in the revised manuscript on **page no 4 and line 19**.

(On page 4, line 19)

“Interfacing both Type A and Type B unit cells in the construction of VPC waveguide domain introduces a sharp kink in the order parameter⁴. Consequently, the topological modes at domain wall of VPC waveguide are commonly referred as ‘kink states’, which exhibits valley-dependent chirality.”

R2.2: In slow light applications, the bandwidth delay product is an important figure of merit. The authors should provide some analysis on this parameter and compare their results with other works in the table.

Our response:

We thank the reviewer for highlighting this point. Following the reviewer’s suggestion, we have included a comprehensive table (Table R1) to compare the normalized bandwidth delay product (NBDP) defined as $n_g \Delta f / f_0$. Here n_g is the group index, Δf is the range of frequency that includes the $\pm 10\%$ variation in n_g and f_0 is the central frequency of the waveguide.

Table R1: Comparing the NBDP with state-of-the-art works.

Platform	NBDP	Topological protection	On-chip compatibility
Silicon rod array ⁶	0.8	No	No
Gyromagnetic photonic crystal ⁷	0.6	No	No
Moire Photonic Crystal ⁸	0.57	No	No
InP defect rods ⁹	0.4	No	No
Photonic Crystals ¹⁰	0.47	No	No
	0.44	No	Yes
	0.5	No	Yes
Valley Photonic Crystal (This work)	0.02	Yes	Yes

Nevertheless, it is important to underline that our present study primarily focuses on unveiling the fundamental mechanism underlying slow light modes in the topological VPC. We have not dedicated substantial effort to enhancing the NDBP in this particular work. Nonetheless, the insights gain from our current work have the potential to guide to design a topological VPC waveguide with near stopped light characteristic, which we intend to explore in our subsequent research endeavours.

R2.3: The topological protection of quantum valley Hall effect is usually quite limited. Although the wave can bend around carefully engineered corners, I am not sure how robust it is against general fabrication imperfections, compared to conventional photonic crystal waveguides.

Our response:

We thank the reviewer for highlighting this point. In response to reviewer's suggestion, we conducted a thorough investigation into the influence of defects on slow light modes within our fabricated VPC waveguide (Type I, $\nu = 3$). To achieve this, we employed two distinct methods:

- We varied the dimensions of triangular holes (Fig. R14a-14c), leading to the introduction of fabrication defects.
- Subsequently, we removed specific triangular holes from the waveguiding path (Fig. R14d-R14f).

To evaluate the consequence of these introduced defects, we computed the corresponding Poynting vectors using COMSOL Multiphysics. The results of these simulations are depicted in Fig. R14. In Fig. R14a-R14c, the dimensions of six triangular holes were varied, ranging from $l_1 = 0.4a/\sqrt{3}$ to $l_1 = 0.55a/\sqrt{3}$, where $a = 260 \mu\text{m}$ represents the periodicity of unit cell. Likewise, in Fig. R14d-R14f, we selectively removed one to three triangular holes. Examining the locus of Poynting vectors in Fig. R14a-R14f reveal that the kink state takes a detour around the defect region while retain its forward and counter-propagating moving waves characteristic even after traversing the region of geometrical imperfections, exhibiting robust transport behaviour.

Nevertheless, to gain more comprehensive understanding and in order to facilitate comparisons with standard photonic crystal waveguide, additional investigations are warranted, which we intend to explore in our subsequent research endeavours.

Fig. R14 Transport of slow light in VPC waveguide through defects (a-c) The locus of Poynting vectors of slow light modes when the defects are introduced by varying the size of the triangular holes from $l_1 = 0.4a/\sqrt{3}$ to $l_1 =$

$0.55a/\sqrt{3}$, where $a = 260 \mu\text{m}$ is the periodicity of unit cell. **(d-f)** The locus of Poynting vectors of slow light modes when the defects are introduced by removing the triangular holes.

In the response to the reviewer's suggestion, we have taken the opportunity to include a description of the effects of defect on slow light modes in VPC waveguide in supplementary information, section S11.

Reviewer #3 (Remarks to the Author):

The topological slow light effect has been demonstrated in valley photonic crystal (VPC) waveguides with a bearded interface. In the manuscript, the authors explain the mechanism of the formation of slow light modes in such bearded-interface VPC waveguides by considering the spatial distribution of magnetic phase vortices around the interface. In addition, the authors experimentally demonstrated the active tuning of group index in a bearded-interface VPC waveguide in the THz regime.

Topological slow-light waveguides are attractive, particularly, from the viewpoint of applications to integrated photonics because they can potentially address some problems in conventional slow-light devices, strong backscattering owing to structural disorders, and huge loss at waveguide bends. Given that, the subject discussed in the manuscript is a very much timely topic and would gather interest in the community. However, I cannot recommend the manuscript for publication in Nature Communications because of the following reasons, at least in its present form.

Our response:

We thank the reviewer for his/her positive comments by highlighting the significance of our work by saying:

"Topological slow-light waveguides are attractive, particularly, from the viewpoint of applications to integrated photonics because they can potentially address some problems in conventional slow-light devices, strong backscattering owing to structural disorders, and huge loss at waveguide bends".

"the subject discussed in the manuscript is a very much timely topic and would gather interest in the community".

Below we address the comments of reviewer in pointwise manner.

R3.1: I can naively understand that the backward-propagating component existing closer to the forward-propagating component makes the group velocity of the valley kink state slower. Is the mechanism unique for bearded-interface VPC waveguides? It is well known that "non-topological" PC waveguides with glide-plane symmetry support slow light modes(*). The authors should discuss the difference in the mechanisms between the two kinds of slow-light waveguides. If the proposed mechanism is unique in topological waveguides, it makes the current work impactful significantly. (*) for example, C. M. Patil et al., "Observation of slow light in glide-symmetric photonic-crystal waveguides," Opt. Express 30, 12565 (2022).

Our response:

We thank the reviewer for highlighting this point. Regarding the reviewer's concern for the existence of backward-propagating (or counter-propagating) component, we want to emphasize that the appearance of counter-propagating (or backward-propagating) Poynting vector arises from two factors: **a) the influence of magnetic phase vortices and b) the stacking of unit cell with a bearded interface, which preserve the glide symmetry** (reflection + translation). This is further highlighted in the supplementary videos (SVa and SVb), which clearly illustrates the existence of counter-propagating waves at bearded interface, while the transport of kink state at zigzag interface VPC waveguide are solely governed by the forward propagating waves.

Regarding the reference cited by reviewer “Observation of slow light in glide-symmetric photonic-crystal waveguides,” *Opt. Express* 30, 12565 (2022)”, we would like to emphasize that even in the cited photonic crystal with glide-symmetry, the underlying mechanism for the observation of slow light mode is the existence of ‘**counter-propagating waves.**’ To demonstrate this, we calculated the projected band diagram of the photonic crystal as shown in Fig. R15a. In the cited reference (*Opt. Express* 30,12565 (2022)), the geometrical parameters of photonic crystal are taken from “*Opt. Mater. Express* 7, 43-51 (2017)”. We reproduced the dispersion curve (Fig. 2b of *Opt. Mater. Express* 7, 43-51 (2017)), as shown in Fig. R15a. Examining the Poynting vector at the slow light frequency (i.e., at band edge) reveals the existence of counter-propagating waves, as depicted in Fig. R15b. Furthermore, to justify the appearance of counter-propagating waves in the photonic crystal we calculate the Berry curvature, shown in Fig. R15c. The nonzero Berry curvature indicates the topological feature thereby warrant the existence of magnetic phase vortices, despite such PC waveguides being labelled as “non-topological”.

In conclusion, our proposed mechanism for the existence of slow light modes in glide symmetry photonic system is universal.

Fig. R15 (a) The dispersion curve of the photonic crystal cited in ref “*Opt. Express* 30, 12565 (2022)” (b) The Poynting vector simulated at band edge frequency, exhibiting the existence of counter-propagating waves (c) Numerically calculated normalized Berry curvature (lower band structure) for the photonic crystal cited in ref “*Opt. Express* 30, 12565 (2022)”.

R3.2: The authors explain that the kink states with large group indexes are formed when the forward and backward propagation modes are coupled strongly. This explanation infers that these are these two modes originally. But they are not identified. Is it appropriate to call them modes?

Our response:

We thank the reviewer for this question. We recognize that the term “counter-propagating modes” might be misleading, as it could imply that the existence of two independent modes and their interaction leading to the slow-light feature. In response to the reviewer’s concern about the existence of two modes (forward and backward), we want to emphasize that the appearance of counter-propagating waves requires the fulfilment of two conditions: a) the existence of magnetic phase vortices and b) the stacking of unit cell with a bearded interface, which preserve the glide symmetry (reflection + translation).

To further emphasize that both the aforementioned conditions are essential for the existence of counter-propagating waves in VPC, we have included a Supplementary Video SVc. This video demonstrates the evolution of the Poynting vector profile, transitioning from “only forward propagating waves” (Fig. R16a and R16b) to “forward and counter-propagating waves” (Fig. R16c and 16d) in the bearded interface waveguide by sweeping across the frequencies for which the group velocity of the topological kink state changes. As the magnetic phase vortices exist for wavevectors ranging from the K point ($k_x = 2\pi/3a$) to the band edge ($k_x = \pi/a$), we observe the appearance of counter-propagating waves from the frequency corresponding to projected K point (i.e., 0.325 THz) to band edge frequency (0.328 THz), as evident in SVc.

Following the reviewer's feedback, we have replaced the term "counter-propagating modes" with "counter-propagating waves" throughout the manuscript to accurately describe the underlying physical phenomenon.

Fig. R16: Evolution of Poynting vector in $\nu = 3$, Type I bearded interface VPC waveguide (a and c) Poynting vector corresponding to the frequency highlighted in 'b' and 'd' (b and d) Transmission spectra through the VPC waveguide. The black dot represents the frequency corresponding to Poynting vectors are plotted in 'a' and 'c'.

In response to reviewer's comment, we have incorporated an additional Supplementary Video, SVC, and updated discussion in the revised manuscript as:

(On page 5, line 5)

“To demonstrate the significance of both these conditions for the existence of counter-propagating waves in the VPC, we have included a Supplementary Video SVC. This video demonstrates the evolution of the Poynting vector profile, transitioning from “only forward propagating waves” to “forward and counter-propagating waves” in the bearded interface VPC waveguide as we sweep across the frequencies corresponding to the K-valley to the band edge. As the magnetic phase vortices exist for wavevectors ranging from the K point ($k_x = 2\pi/3a$) to the band edge ($k_x = \pi/a$), we observe the appearance of counter-propagating waves from the frequency corresponding to the projected K point to the band edge frequency, as clearly evident in SVC.”

R3.3: The authors did not provide enough explanation about the assignment of topological guiding mode. In Type I with $\nu = 3$, where smaller holes are facing the interface, the lower-frequency branch will be a topological mode (according to [27]). On the other hand, when $\nu = 6$ and 9, the higher-frequency branch is assigned to a topological band. How did the authors know which branch of the in-gap modes is topological? Why does the change mentioned above happen?

Our response:

We thank the reviewer for highlighting this out. We concur with the reviewer's insight that, for $\nu = 3$ and Type I interface, the introduction of sharp bends could serve as a mean to discern the topological mode, as exemplified in Fig. R8. However, to accurately identify the topological and trivial modes within the bearded interface VPC waveguide, it is appropriate to observe the counter-propagating waves. For example, in case of $\nu = 3$, Type II bearded interface VPC waveguide, we identified the topological and trivial modes by examining the Poynting vectors at K- valley, as depicted in Fig. R17a and R17b. It is evident that for $\nu = 3$ and Type II bearded interface, the lower band is the topological band as there

exists counter-propagating waves (Fig. R17a) whereas the upper band signifies the trivial band as evidenced by the absence of counter-propagating waves in the Poynting vector (Fig. R17b).

Fig. R17 Identification of topological and trivial mode in $\nu = 3$, Type II bearded VPC waveguide (a and b)
The locus of Poynting vectors corresponding to lower and upper band, respectively.

Similarly, we extended this procedure for $\nu = 9$ Type I and Type II bearded interface VPC waveguide as depicted in Fig. R11 and R12, respectively.

Fig. R18a and R18d show the projected band diagram for $\nu = 9$, Type I bearded interface, where the topological and trivial modes are indicated by solid and dashed line, respectively. Correspondingly, Fig. R18b and R18e show the transmission spectra through the $\nu = 9$, Type I VPC waveguide, where the combination of solid and dashed lines highlights the transmission corresponding to topological and trivial modes, respectively. Upon careful analysis of the Poynting vector linked to the topological modes, we observe the emergence of counter-propagating waves (depicted as blue arrows). These counter-propagating waves are a consequence of magnetic phase vortices and serve as an indicator of the topological characteristics, as shown in Fig. R18c and R18f.

Fig. R18 Identification of topological and trivial mode in $\nu = 9$, Type I bearded VPC waveguide (a and d) The projected band diagram (b and e) Simulated transmission spectra. The red dot marks the trivial and topological modes for which Poynting vectors are simulated in 'c' and 'f' (c and f) The locus of Poynting vectors corresponding to trivial and topological modes, respectively.

Fig. R19a and R19d show the projected band diagram for $\nu = 9$, Type II bearded interface VPC waveguide, where the trivial and topological modes are highlighted by solid and dashed lines, respectively. Fig. R19b and R19e depict the transmission spectra, where the hybridization of solid and dash lines highlights the contribution corresponding to topological and trivial modes, respectively. Likewise, examining the locus of Poynting vectors corresponding to topological mode reveal the presence of counter-propagating waves (Fig. R19c). However, the trivial mode lacks such counter-propagating waves (Fig. R19f).

Fig. R19 Identification of topological and trivial mode in $\nu = 9$, Type II bearded VPC waveguide (a and d) The projected band diagram (b and e) Simulated transmission spectra. The blue dot marks the trivial and

topological modes for which Poynting vectors are simulated in 'c' and 'f' (**c and f**) The locus of Poynting vectors corresponding to topological and trivial modes, respectively.

In the response to the reviewer's suggestion, we have taken the opportunity to include a description on the identification of topological and trivial modes. Furthermore, we have included illustrative videos SVd and SVe and have updated the supplementary information, section S1. The revised manuscript now incorporates the following updated text:

(On page 7, line 20)

"It is noteworthy that, to precisely the distinguish between the topological and trivial modes within the bearded interface VPC waveguide, it is appropriate to observe the presence of counter-propagating waves (see detailed discussion in SI, section S1)."

R3.4: The mechanism for the change in the group index under photoexcitation seems to be explained differently from that in the first half of the manuscript. The authors explain that the extra detour of light, not the change in the distance between the forward- and backward-propagating components, causes the change in the group index. It might be my misunderstanding of the author's intention. But I feel the authors need to explicitly explain whether the tuning mechanism is the same or not with the mechanism discussed in the first half of the manuscript so that the readers will not misunderstand. To directly show that the distance between the forward- and backward-propagating components changes under the photoexcitation, the authors need to show the distributions of the Poynting vector of the eigenmode when non-zero conductivity.

Our response:

We thank the reviewer for pointing this out. Since, the interaction of forward and counter-propagating waves is the underlying reason behind the appearance of slow light in bearded interface VPC waveguide. We demonstrated that by **changing the coupling strength between the forward and counter-propagating waves**, the slowness in VPC waveguide can be tuned. The coupling strength can be tuned by the following means:

- a. **By engineering the bandgap (which is shown in the first part of manuscript):** In this approach, altering the bandgap allows us to tune the transverse decay length of the topological kink modes perpendicular to the interface, which affects the coupling strength between forward and counter-propagating waves. This concept is illustrated in Fig.2 of the manuscript, where we achieve the bandgap tunability by changing the shape of air holes from triangle to nonagon. To further emphasize this, we performed an extensive investigation of the bandgap and group velocity (v_g). For this purpose, we fixed the air hole shape to be hexagon ($\nu = 6$) and varied the air hole radii (r_1, r_2) for hexagonal air holes ($\nu = 6$) of period $a = 260 \mu m$. Fig. R20a presents the bandgap of VPCs, with colormap representing the strength of the bandgap. We selected the unit cells with increasing bandgap, as marked in Fig. R20a. The specific radii (r_1, r_2) of these chosen unit cells are as follows: $(0, 0.79a/\sqrt{3})$, $(0.22a/\sqrt{3}, 0.78a/\sqrt{3})$, $(0.29a/\sqrt{3}, 0.75a/\sqrt{3})$, $(0.33a/\sqrt{3}, 0.71a/\sqrt{3})$ and $(0.37a/\sqrt{3}, 0.67a/\sqrt{3})$, with corresponding bandgap (Δf) 166 GHz, 141 GHz, 115 GHz, 84 GHz and 61 GHz, respectively.

Increasing the bandgap of VPC leads to increased transverse decay of the topological modes perpendicular to the interface, resulting in the enhancement of group velocity (v_g) of the kink states due to weak coupling between forward and counter-propagating waves. To further elaborate, we plotted the group velocity extracted at the Brillouin zone edge ($v_g^{\pi/a}$ where $k_x = \pi/a$) for Type I and Type II bearded interfaces in Fig. R20b. In the case of Type I bearded interface, the proximity of the adjacent counter-propagating wavefront to the interface causes the slowness (or $v_g^{\pi/a}$) to remain relatively unchanged, even with an increase in bandgap, as shown by red solid line in Fig. R20b. Conversely, for Type II bearded interface, increasing the bandgap reduces the spatial extent of the topological modes perpendicular to the interface, leading to weak coupling between the forward and adjacent counter-propagating waves.

Consequently, $v_g^{\pi/a}$ increases with the increase in bandgap as shown by blue solid line in Fig. R20b. This is further corroborated through the Poynting vector (\vec{S}) plots depicted in Fig. R20d-20h, where we observe gradual decrease in the strength of the counter-propagating waves with an increase in bandgap (from left to right), which results in an increase in $v_g^{\pi/a}$.

Fig. R20: Tuning the slowness of kink state in VPC waveguide via bandgap engineering. (a) Bandgap of VPCs as a function of air hole radii (r_1, r_2) for hexagonal air holes ($v = 6$) of period $a = 260 \mu\text{m}$. The colormap represents the strength of bandgap. Markers show the selected unit cells with increasing bandgap. The specific radii (r_1, r_2) of these chosen unit cells are as follows: $(0, 0.79a/\sqrt{3})$, $(0.22a/\sqrt{3}, 0.78a/\sqrt{3})$, $(0.29a/\sqrt{3}, 0.75a/\sqrt{3})$, $(0.33a/\sqrt{3}, 0.71a/\sqrt{3})$ and $(0.37a/\sqrt{3}, 0.67a/\sqrt{3})$, with corresponding bandgap (Δf) 166 GHz, 141 GHz, 115 GHz, 84 GHz and 61 GHz, respectively. (b) The band edge group velocity ($v_g^{\pi/a}$) is calculated for Type I (red solid line) and Type II (blue solid line) bearded interface VPC waveguides, shown on the left Y-axis. The bandgap corresponding to selected unit cells is shown on the right Y-axis (c-g) The locus of the Poynting vector is extracted at $k_x = \pi/a$, where the bandgap increases from 61 GHz to 166 GHz from left to right.

b. By actively modulating the strength of the forward and counter-propagating waves via photoexcitation: This is an active way to tune the slowness of the kink edge in VPC waveguide, which is shown in the second half of the manuscript. Photoexciting the silicon above the bandgap using a wavelength of $\lambda = 532 \text{ nm}$ generates free carrier in the silicon that induces loss, consequently acting as a local ‘non-Hermitian defect’. Initially, increasing the power of optical pump (or the conductivity of silicon in the simulation) attenuates the counter-propagating waves, resulting in a decrease in group delay, shown in Fig. R5. However, further increasing the pump power enables the kink state to take detour around the circular region (pump spot) and maintain its forward and counter-propagating waves after traversing the circular region (Fig. R5e), resulting in an increase in group delay, as depicted in Fig. R5b.

To provide a clear explanation of the tuning mechanism of slow light in VPC waveguide, we have expanded the discussion in the revised manuscript and made modifications in the figures (Fig 3 and Fig 4). The updated texts are as follow:

(On page 9, line 17)

“To further emphasize the effect of the band gap on v_g , we conducted an extensive investigation by fixing the air hole shape to be a hexagon ($v = 6$) and varied the air hole radii (r_1, r_2) of period $a = 260 \mu\text{m}$. Fig. 3a presents the band gap of VPCs, with the colormap representing the strength of the

band gap. We selected the unit cells with increasing band gap, as marked in Fig. 3a. The specific radii (r_1, r_2) of these chosen unit cells are as follows: $(0, 0.79a/\sqrt{3})$, $(0.22a/\sqrt{3}, 0.78a/\sqrt{3})$, $(0.29a/\sqrt{3}, 0.75a/\sqrt{3})$, $(0.33a/\sqrt{3}, 0.71a/\sqrt{3})$ and $(0.37a/\sqrt{3}, 0.67a/\sqrt{3})$, with corresponding band gap (Δf) 166 GHz, 141 GHz, 115 GHz, 84 GHz and 61 GHz, respectively. The projected band diagrams for these chosen unit cells are depicted in Fig. 3d-3h, showing a clear increase in the band gap from left to right. Fig. 3b shows the group velocity extracted at Brillouin zone edge ($v_g^{\pi/a}$ where $k_x = \pi/a$) for Type I and Type II bearded interface VPC waveguides. For Type I bearded interface, the proximity of the counter-propagating wave to the interface causes the slowness (or $v_g^{\pi/a}$) to remain relatively unchanged, even with an increase in band gap, as shown by red solid line in Fig. 3b. In contrast, for Type II bearded interface, increasing the band gap reduces the spatial extent of the topological modes perpendicular to the interface, leading to weak coupling between the forward and adjacent counter-propagating waves. Consequently, $v_g^{\pi/a}$ increases with the increase in band gap as shown by blue solid line in Fig. 3b. This observation is further supported by the Poynting vector (\vec{S}) plots depicted in Fig. 3i-3m, where we observe gradual decrease in the strength of the counter-propagating waves with an increase in band gap (from left to right) for Type II interface, which results in an increase in $v_g^{\pi/a}$.

Moreover, to demonstrate the universality of the underlying mechanism of slow light in bearded interface VPC waveguide, which arises from the interaction between forward and counter-propagating waves, we calculated the group velocity at the projected K valley i.e., ($v_g^{2\pi/3a}$ where $k_x = 2\pi/3a$), as shown in Fig. 3c. Similarly, there is no correlation between $v_g^{2\pi/3a}$ and the band gap for Type I bearded interface while $v_g^{2\pi/3a}$ increases with the band gap for Type II interface, which is also supported by the Poynting vector plots (Fig. S4)."

R3.5: I could not understand the relationship between the inset of Fig. 3 c) and Fig. 3 d). In the inset of Fig. 3 c), the group index has a peak at the conductivity of ~ 50 S/m. Does this conductivity correspond to kappa of 0.5 in Fig. 3 d)? Why are the group indexes between the two plots different largely?

Our response:

In the response to the reviewer's comment and to avoid the confusion between the previous Fig. 3c and 3d, we chose to utilize the group delay (GD) rather than the group index (n_g). This is because group index represents a global parameter and does not accurately reflect the localized changes in the refractive index induced by optical pumping. We have replaced the PWE simulations (previous Fig. 3d) with 3D CST simulations which allow us to directly compute the group delay. As per Figure R21, our experimental and simulated group delay is in good agreement.

Moreover, to provide comprehensive discussion of the active tuning of the group index, we have elaborated on the discussion of PWE in the **supplementary information, section S8**. This additional information aims to enrich the understanding of the approach used for the active tuning of the group index in our study. In regard to the reviewer's concern about the disparity in group indexes between the COMSOL simulation (previous Fig 3c) and PWE calculation (previous Fig. 3d), we would like to point out the following reasons for the differences:

- The eigenvalue calculation done in COMSOL (previous Fig. 3c) was performed in a 3D geometry, while the PWE was employed in a 2D geometry.
- Additionally, the conductivity value used in the COMSOL eigenvalue solver does not have one to one correspondence with the kappa (imaginary part of refractive index of silicon) in the PWE calculation.

In summary, to prevent any confusion we chose to utilize the group delay (GD) rather than the group index (n_g) and replaced the previous Fig. 3c with the group delay plot extracted using 3D simulation to accurately emulate the experimental scenario. We have made the necessary modifications to the **figure and updated it in the revised manuscript as Fig. 4**.

The updated discussion in the revised manuscript is marked as:

(From page 12, line 19 to page 13, line 25)

“We record the GD defined as $GD = -\frac{d\phi}{d\omega}$, (where ϕ and ω are phase and angular frequency of the transmitted wave, respectively) from the VPC waveguide chip using a vector network analyser (VNA) based experimental setup (see SI, section S5). We photoexcite the domain wall of VPC waveguide chip (see Fig. 3a) with a continuous wave laser of wavelength 532 nm to tune the slowness. The optical beam spot (diameter ≈ 0.5 mm) (see SI, Fig. S6 and S7) was kept small to ensure uniform illumination of few unit cells. Fig. 3b shows the variation of GD as a function of optical pump power. To accurately replicate the experimental conditions, we performed 3D simulation in CST Microwave Studio, as thoroughly explained in SI, section S7. Given that the optical beam spot in the experiment is focused on only a few unit cells (Fig. 3a), we emulate the effects of optical pumping by defining a circular region spanning a few unit cells (highlighted in red in Fig. S8) at the centre of the bearded interface of the VPC waveguide. Within this region, the conductivity of silicon is varied, which is equivalent to changing the optical pump power in experiment. Fig. 3b shows a comparison between the experimental and simulated GD curves, which shows good agreement in the variation of GD. Importantly, we observe an identical relative change (i.e., difference between maximum and minimum GD) in GD values for both the experiment and simulation, which is around 11%. The slight variations in GD values can be attributed to slight experimental misalignment and variations in the thickness (± 25 μm) of the fabricated VPC waveguide.

To understand the behaviour of GD as a function of optical pump power (or Si conductivity in simulation), we computed Poynting vector plots from 3D simulations. Fig. 3c-3e show the locus of Poynting vector calculated at the conductivity values marked in the inset of Fig. 3b, where the black circle encloses the region with variable Si conductivity. For $\sigma = 0$ S/m, the transport of kink state is accompanied by forward and counter-propagating waves, as depicted in Fig. 3c. As the conductivity of Si increases, free carrier densities increase and imparts losses, as a consequent the photoactive region acts as a non-Hermitian defect that attenuates the forward propagating waves, resulting in a decrease in GD, as shown in Fig. 3d. With further increase in the conductivity of Si, the kink state takes a detour around the circular region and retain its forward and counter-propagating waves after traversing the circular region (Fig. 3e), resulting an increase in GD, as evident in Fig. 3b. To ensure the consistency of our results, we conducted photoexcitation measurements multiple times by gradually increasing and decreasing the optical pump power. The results are depicted in Fig. S15, showing no variation in the measured GD values. This observation rules out the possibility of any hysteresis-type behaviour and validate the reproducibility in our experiment.”

In addition, the extended discussion on simulation is presented in the supplementary information, section S7 and S9.

R3.6: It was unclear to me how the authors estimated the group index under photoexcitation. I suppose they used Eq. (2) with $L = 8.97$ mm. If yes, the estimated group index should be understood as an “effective” or “averaged” one since the change in the group index only occurs within one- or two-unit cells as the authors discussed in LL265-268. Related to this point, why could the authors the group indexes similar to the experimentally obtained values by using the PWE method?

Our response:

We thank the reviewer for providing detailed feedback. We agree with the reviewer’s observation that the group index (n_g) is not a suitable metric to characterize the localized changes induced by photoexcitation. Consequently, the previously estimated group index using equation 2 (in the previous version of manuscript) does not accurately capture the effective or averaged change in the VPC waveguide upon photoexcitation.

To address the disparity between the experimental results involving optical pumping and simulations, we have chosen to compare the group delay (GD) instead of the group index (n_g), for the following reasons:

- GD directly quantifies the time delay experienced by signal or electromagnetic waves as they propagate through the waveguide, providing a more accurate measure of temporal changes induced by localized changes. On the other hand, group index (n_g) is a measure of the refractive index of a waveguide, considering the dispersion of the waveguide.
- In the case of photoexcitation, the optical pump spot spans up to a few unit cells, inducing free carrier locally, which in turn effectively leads to changes in the refractive index of these few unit cells. As the group index (n_g) is a global parameter that encapsulates the effective refractive index of the entire waveguide, n_g is not a good metric to characterize the localized changes induced by photoexcitation.

Fig. R21: The experimental GD is plotted as a function of optical pump power. The GD values are obtained using a vector network analyzer (VNA) based setup. The inset graph illustrates the simulated GD against the conductivity of silicon, which shows good agreement with experimental results. To replicate the experimental conditions, a 3D approach is utilized in CST Microwave Studio.

In the response to the reviewer’s comment, we have replaced the PWE simulations (previous Fig. 3d) with 3D CST simulations (now Fig. 3c) which allow us to directly compute the group delay. As depicted in Fig. R21, our experimental and simulated group delay exhibit good agreement, providing a more accurate representation of experimental scenario.

The updated texts in the revised manuscript are as follow:

(On page 13, line 11)

“To understand the behaviour of GD as a function of optical pump power (or Si conductivity in simulation), we computed Poynting vector plots from 3D simulations. Fig. 3c-3e show the locus of Poynting vector calculated at the conductivity values marked in the inset of Fig. 3b, where the black circle encloses the region with variable Si conductivity. For $\sigma = 0$ S/m, the transport of kink state is accompanied by forward and counter-propagating waves, as depicted in Fig. 3c. As the conductivity of Si increases, free carrier densities increase and imparts losses, as a consequent the photoactive region acts as a non-Hermitian defect that attenuates the forward propagating waves, resulting in a decrease in GD, as shown in Fig.3d. With further increase in the conductivity of Si, the kink state takes a detour around the circular region and retain its forward and counter-propagating waves after traversing the circular region (Fig. 3e), resulting an increase in GD, as evident in Fig. 3b. To ensure the consistency of our results, we conducted photoexcitation measurements multiple times by gradually increasing and decreasing the optical pump power. The results are depicted in Fig. S15, showing no variation in the measured GD values. This observation rules out the possibility of any hysteresis-type behaviour and validate the reproducibility in our experiment.”

In addition, the extended discussion on simulation is presented in the supplementary information, section S7 and S9.

References

1. Hammer, M. & Ivanova, O. V. Effective index approximations of photonic crystal slabs: a 2-to-1-D assessment. *Opt Quant Electron* **41**, 267–283 (2009).
2. Tan, Y. J., Wang, W., Kumar, A. & Singh, R. Interfacial topological photonics: broadband silicon waveguides for THz 6G communication and beyond. *Opt. Express*, *OE* **30**, 33035–33047 (2022).
3. Gao, F. *et al.* Topologically protected refraction of robust kink states in valley photonic crystals. *Nat. Phys.* **14**, 140–144 (2018).
4. Yao, W., Yang, S. A. & Niu, Q. Edge States in Graphene: From Gapped Flat-Band to Gapless Chiral Modes. *Phys. Rev. Lett.* **102**, 096801 (2009).
5. Yang, Y. *et al.* Terahertz topological photonics for on-chip communication. *Nature Photonics* **14**, 446–451 (2020).
6. Wu, H., Han, S., Li, F. & Yang, Z. Slow light with high normalized delay–bandwidth product in organic photonic crystal coupled-cavity waveguide. *Appl. Opt.*, *AO* **59**, 642–647 (2020).
7. Liu, Y. & Jiang, C. Dispersionless one-way slow wave with large delay bandwidth product at the edge of gyromagnetic photonic crystal. *Int. J. Mod. Phys. B* **34**, 2050086 (2020).
8. Nasidi, I., Hao, R., Chen, J., Li, E. & Jin, S. Photonic Moiré lattice waveguide with a large slow light bandwidth and delay-bandwidth product. *Appl. Opt.*, *AO* **61**, 5776–5781 (2022).
9. Ma, Y., Wu, R. & Li, L. Research on slow light transmission with wide bandwidth and large normalized delay bandwidth product. *Optoelectron. Lett.* **17**, 407–411 (2021).
10. Pourmand, M., Karimkhani, A. & Nazari, F. Wideband and low-dispersion engineered slow light using liquid infiltration of a modified photonic crystal waveguide. *Appl. Opt.*, *AO* **55**, 10060–10066 (2016).

REVIEWER COMMENTS

Reviewer #1 (Remarks to the Author):

The authors have addressed most of my technical points and I think the manuscript now provides a much better justification of the mechanism for the origin of slow light (thanks to the simulations as a function of size disparity) and is also more faithful in the claims of what the active tuning implies. I appreciate the efforts made. However, as I already anticipated on the first review round, I still do not find that the manuscript has the impact that grants publication in Nature Communications. Nevertheless, I understand that sometimes impact is a hard thing to gauge, and I leave it upon the Editor to decide that.

The reasons I believe the updated manuscript still does not deserve publication are the following:

1-The active control on the group delay is based on a completely different approach than the origin of the slowness itself (despite what the authors seem to imply on answer R3.4 to Reviewer 3). It is based on creating a defect that the mode “bypasses” and not on controlling the coupling between forward and backward “waves”. I would encourage the authors to try to find all points in the manuscript where one could infer the connection between the two aspects of the manuscript and clarify that they are not connected. Notably, this is implied in the abstract. Also, I do not believe that the photoexcitation experiments (with the current presented data, see below on the technical points) manifest any beneficial aspect of the topological mode for technological applications.

2- The findings in response to the comment R3.1 of Reviewer 3 imply that either the main key aspect here is the glide plane symmetry or that the waveguides in [C. M. Patil et al., “Observation of slow light in glide-symmetric photonic-crystal waveguides,” *Opt. Express* 30, 12565 (2022)] are topological, as the authors seem to imply. I am open to accept that many more structures that we believe are topological, but those waveguides are clearly not topological in the conventional sense, i.e., they are a line defect and not an interface. I believe that one of the reasons why this manuscript is considered for publication in Nature Communications is the fact that the waveguides are topological, but if the mechanism is not completely exclusive to topological waveguides and it mainly results from the glide plane symmetry, I have my doubts it would be considered for publication. This, on a lesser negative tone, is also highlighted by Reviewer 3 on point R3.1. Can the authors show a glide plane symmetric waveguide that does not exhibit and is not topological? In conclusion, the “topologicalness” to the waveguides might be incidental, somehow.

3-Even if the authors have attempted at addressing my concerns about disorder with new simulations, these are point-like defects in an otherwise perfect system. I do not believe this represents real world disorder and I think the authors have not fully considered the recent literature on the effect of

fabrication disorder for exactly these structures, e.g., the previously mentioned references [C. Rosiek et al, Nature Photonics 17, 386 (2023)] and [E. Verhagen, Proceedings Volume PC12196, Active Photonic Platforms 2022, SPIE 2022; <https://doi.org/10.1117/12.2632855>]. If topological waveguides are not to be protected against fabrication disorder is the mechanism of the slow light in these waveguides enough to grant publication in Nature Communications? I also want to highlight that the authors responded to my comment on the effect of disorder by mentioning that the operating wavelength is three orders of magnitude longer than in the telecom-band valley-Hall waveguides of the references, but the effect of disorder would be the same if the disorder is of several microns as they themselves suggest. What matters is the size of disorder relative to the pitch and in the waveguides of the references that would also be three orders of magnitude smaller (few nanometers at maximum). This can be easily understood from perturbation theory expressions to the loss rates [see for example S. Hughes, L. Ramunno, Jeff F. Young, and J. E. Sipe, Phys. Rev. Lett. 94, 033903 (2005)]. In conclusion, what would make slow light in these waveguides appealing for technological applications in integrated photonics, i.e., their supposed topological protection, probably does not manifest in their structures. Maybe a discussion on the disorder levels of the structures (roughness) or loss measurements could be helpful.

In addition to these main arguments, I would like to point a few more aspects:

1-The authors have extended on these valuable simulations about the photoexcitation experiment. In relation to Fig. S13, the authors say: “it becomes apparent that the kink state takes a detour around the circular region and retain its forward and counter-propagating waves after traversing the circular region. In contrast, for the trivial mode, there is a significant scattering and back reflection with an increase in silicon conductivity”. I sort of understand why the authors say that the mode takes a detour, but I want to highlight that the fact that it retains the forward and counter-propagating waves after it is not very relevant. It is a single-moded waveguide so only that mode can potentially propagate after the defect (as also happens for the trivial mode). I think plots of simulated transmission as a function of the conductivity and experimental transmission as a function of the pump power are needed to see how efficiently the topological mode bypasses those defects. The same applies to the simulations of the artificial disorder (Fig. S16). If the “active control” of the group delay incurs large losses, then it is not so interesting.

2- Whenever the authors talk about topological protection, they should specify which topological waveguides they are talking about (time-reversal-symmetric or not) and against which type of defects.

3- The authors say: “Compared to conventional PC platforms, the vortex-driven slow light effect in VPCs offer significant flexibility to engineer the functionality of topological slow light photonic devices.” It is quite unclear to me why the authors say this and if it is justified by their work.

4- Fig. 1a is still very poorly rendered. The arrows are extremely pixelized upon zooming.

5- Regarding the distinction between the topological and trivial mode, which has been tackled more in detail in the updated version of the manuscript, the authors say: "It is noteworthy that, to precisely distinguish between the topological and trivial modes within the bearded interface VPC waveguide, it is appropriate to observe the presence of counter-propagating waves (see detailed discussion in SI, section S1)." For me this is quite a tautological argument, given that this is what they are actually trying to demonstrate in the manuscript. I think another argument should be provided, like a simulation of the transmission across Z-shaped bends.

6- I feel like Fig. 2 could now be in the SI, since Fig. 3 serves roughly the same purpose but gives much better arguments for the conclusion that the authors make.

7- Page 12, Lines 13-18: I think the authors should drop this discussion about the group index. This distinction between the local variation and the global variation was brought up during the review and is now corrected. The justification of why one should evaluate GD instead of n_g feels unnecessary now and may only add confusion.

8- In Table 1, the column with "Topological protection" is confusing. Do the authors imply that they have shown topological protection? This connects to my previous point number 2 here.

9- The "active" in the title is confusing. There is nothing active about the slow light. I believe the only way the word active could come in would be in connection to the words group delay, e.g., "active group delay control" or something similar.

Reviewer #2 (Remarks to the Author):

The authors have addressed all the concerns in my last report. I am happy to recommend it for publications in Nature Communications in its present form.

Reviewer #3 (Remarks to the Author):

I appreciate the authors' great efforts in responding to my comments. I am satisfied with their response except for the assignment of trivial and topological bands. The authors assigned the modes that have back-propagation modes as topological. Naively, it is expected that those modes show high transmittance over sharp bends. Before recommending the manuscript for publication, I request the authors to demonstrate this feature by showing the computed transmission spectra in the structures with sharp bends. (I suppose that the transmission spectra in Figs. S1. 3 and 4 are the ones for straight waveguides). It is informative to point out the one-to-one correspondence between the waveguiding properties against sharp turns, which is widely known as a feature of topological modes, and the presence of the counter-propagating waves. In addition, it is very interesting that the topological band in the Type 1 structure changes from the lower-frequency band to the higher-frequency band when v changes from 3 to 6 and 9. Could the authors discuss the reason for the change?

Response Letter to Reviewers

We thank all the reviewers for their constructive comments and input on our work. The comments have helped us to enrich the content of the manuscript.

In the subsequent text, the comments from the reviewer are presented in 'black' text, followed by our comprehensive response highlighted in 'blue'. Taking into consideration the reviewer's comment, the revisions incorporated in the revised manuscript and supplementary information are highlighted in yellow. The modified texts in the revised manuscript are posted here in the *italic* font.

Reviewer #1 (Remarks to the Author):

The authors have addressed most of my technical points and I think the manuscript now provides a much better justification of the mechanism for the origin of slow light (thanks to the simulations as a function of size disparity) and is also more faithful in the claims of what the active tuning implies. I appreciate the efforts made. However, as I already anticipated on the first review round, I still do not find that the manuscript has the impact that grants publication in Nature Communications. Nevertheless, I understand that sometimes impact is a hard thing to gauge, and I leave it upon the Editor to decide that.

Our response:

We thank the reviewer for applauding our results by saying "*I think the manuscript now provides a much better justification of the mechanism for the origin of slow light and is also more faithful in the claims of what the active tuning implies*". Below we address the updated concern of the reviewer in pointwise manner.

The reasons I believe the updated manuscript still does not deserve publication are the following:

R1.1: The active control on the group delay is based on a completely different approach than the origin of the slowness itself (despite what the authors seem to imply on answer R3.4 to Reviewer 3). It is based on creating a defect that the mode "bypasses" and not on controlling the coupling between forward and backward "waves". I would encourage the authors to try to find all points in the manuscript where one could infer the connection between the two aspects of the manuscript and clarify that they are not connected. Notably, this is implied in the abstract. Also, I do not believe that the photoexcitation experiments (with the current presented data, see below on the technical points) manifest any beneficial aspect of the topological mode for technological applications.

Our response:

We thank the reviewer for highlighting this point. Regarding the confusion between passive and active ways to tune the slow light in VPC waveguide, we clearly want to emphasize that the mechanism of active tuning still requires the existence of counter-propagating waves as clearly presented in the change of the Poynting vector profile as a function of non-Hermitian defect conductivity, as depicted in Fig. 4 of the manuscript. Following the reviewer's suggestion, we have modified the text in the revised manuscript to clearly layout the connection between passive and active tuning of the topological slow light.

The updated texts in the revised manuscript are as follow:

Page 12 and line 11

“As discussed above, the presence of counter-propagating waves in the vicinity of the bearded interface, facilitates the slow light effect in VPC waveguide. Therefore, we sought to actively control the spatial distribution of these counter-propagating waves through photoexcitation, providing a way to tune the slow light effect in VPC waveguide. It is worth noting that, akin to tuning the slowness of kink state through band gap engineering (as shown in Fig. 2), photoexcitation also tunes the coupling between forward and counter-propagating waves by modulating the terahertz waves through free carriers.”

Page 14 and line 17

“The interaction between forward and counter-propagating waves gives rise to the slow light effect in VPC waveguides, which can be actively and passively controlled by adjusting the coupling strength between them.”

Moreover, regarding the reviewer’s remark on the technological benefits of active topological slow light effect, we highlight that to devise a terahertz photonic integrated circuit it is imperative to have key functionalities such as switches, filters, and modulators. To achieve these, active control of electromagnetic waves on a chip is vital. We present a perspective view of the topological integrated photonic chip in Fig. R3, to showcase the application outlook and importance of active slow light effect.

R1.2: The findings in response to the comment R3.1 of Reviewer 3 imply that either the main key aspect here is the glide plane symmetry or that the waveguides in [C. M. Patil et al., “Observation of slow light in glide-symmetric photonic-crystal waveguides,” *Opt. Express* 30, 12565 (2022)] are topological, as the authors seem to imply. I am open to accept that many more structures that we believe are topological, but those waveguides are clearly not topological in the conventional sense, i.e., they are a line defect and not an interface. I believe that one of the reasons why this manuscript is considered for publication in *Nature Communications* is the fact that the waveguides are topological, but if the mechanism is not completely exclusive to topological waveguides and it mainly results from the glide plane symmetry, I have my doubts it would be considered for publication. This, on a lesser negative tone, is also highlighted by Reviewer 3 on point R3.1. Can the authors show a glide plane symmetric waveguide that does not exhibit and is not topological? In conclusion, the “topologicalness” to the waveguides might be incidental, somehow.

Our response:

We thank the reviewer for the comments. However, we want to clarify that the glide-symmetry (GS) waveguide design presented in “*Opt. Express* 30, 12565 (2022)” involves a series of transformations and structural modifications to the conventional W1 photonic-crystal waveguide. For instance, the photonic-crystal cladding of W1 waveguide is shifted by a half period to achieve glide symmetry in GS waveguide. It is worth noting that the cladding of GS waveguide is made of triangular lattice photonic crystal, which exhibits magnetic phase vortices and non-zero Berry curvature, as illustrated in Fig. R1a and R1b. Essentially, the GS waveguide represents a structurally modified Type I VPC waveguide with circular holes of radius $r_1 = 0.5a/\sqrt{3}$ and $r_2 = 0$, where ‘ a ’ is the lattice periodicity, with an additional separation of $\Delta y = \frac{a}{2\sqrt{3}}$, as depicted in Fig. R1d.

Note that in VPC, the circular hole is centered at the centroid of each triangular half of the rhombus/diamond unit cell, while for conventional PC, the circular hole is positioned at the center of the unit cell. Therefore, the additional separation of $\Delta y = \frac{a}{2\sqrt{3}}$ at the interface of Type I VPC waveguide results in a glide symmetric W1 waveguide, where a single row of air holes is removed to create a separation between top and bottom cladding. This separation between top and bottom cladding serves as a line defect, as pointed out by the reviewer as well. It is important to highlight that we are not claiming the GS waveguide presented in *Opt. Express* 30, 12565 (2022) is topological, rather we highlight that the cladding of GS waveguide fulfills both criteria for the existence of counter-propagating waves, namely: **(a)** presence of magnetic phase vortices with non-zero Berry curvature and **(b)** glide-plane symmetry, resulting in the transport of electromagnetic modes accompanied by counter-propagating waves, as shown in Fig. R1. The introduction of line defect reduces the coupling strength between the top and bottom cladding, which subsequently diminishes the topological robustness of GS waveguide.

Fig. R1. Topological characteristics of glide symmetric (GS) waveguide presented in Opt. Express 30, 12565 (2022) (a) Unit cell of the GS waveguide, exhibiting the magnetic phase vortex (b) Numerically calculated Berry curvature (c) Projected strip of the Type I VPC waveguide overlaid with the x-component of Poynting vector (S_x). The locus of Poynting vector reveals the counter-propagating waves. (d) Projected strip of the standard W1 waveguide with glide symmetry (GS waveguide), which is also constructed by separating the interface of Type I waveguide by $\frac{a}{2\sqrt{3}}$. (e) Defect waveguide constructed by separating the triangular cladding by $\Delta y = \frac{a}{2\sqrt{3}} + \frac{a\sqrt{3}}{4}$.

Furthermore, to underscore our points and to address the reviewer's concern that "Can the authors show a glide plane symmetric waveguide that does not exhibit slow light and is not topological", we examined the locus of Poynting vector for a waveguide composed of all-dielectric C4 photonic crystal, as presented in Fig. R2a and R2b. In the case of the C4 photonic crystal, magnetic phase vortices do not exist due to zero Berry curvature. Consequently, even when stacking the unit cell to form glide-plane symmetry, it does not warrant the existence of counter-propagating waves with in-plane confinement, as shown in Fig. R2a and R2b. Here, we examined all the eigenmodes from $k_x = 0$ to $k_x = \pi/a$ within the band gap and presented the most similar case terms of Poynting vectors in Fig. R2a and R2b, where there is a clear observation of direct energy flow into the bulk (y -direction). It is evident that no counter-propagating waves are excited in the waveguide at the frequency of that eigenmode presented in Fig. R2a and R2b (see Fig. R2d).

Furthermore, to emphasize that all-dielectric C4 photonic crystal is not topological, we assessed the transmission through a waveguide comprising two sharp bends (90-degrees). It is important to note that geometrically it is impossible to design a 90-degree bend in C4 photonic crystal with glide symmetry. Consequently, we constructed a 90-degree bend with mirror symmetry. Fig. R2c depict the transmission through straight (C4 + glide symmetry) and bend domain walls. Due to the absence of topological protection, transmission through sharp bends (red solid line) is poor, as compared to straight domain wall (blue solid line). Fig. R2d show the excited electric field profile of the straight domain corresponding to the frequency of eigenmodes in R2a and R2b, showing the absence of counter-propagating waves.

Fig. R2. Analysis of C4 photonic crystal with glide symmetry (a,b) Projected strip of C4 photonic crystal with glide symmetry overlaid with x and y components of the Poynting vector (S), respectively. **(c)** Transmission spectra corresponding to straight (blue solid line) and 90-degree bend (red solid line) C4 photonic crystal waveguide. **(d)** x component of the Poynting vector for straight waveguide with glide symmetry.

R1.3: Even if the authors have attempted at addressing my concerns about disorder with new simulations, these are point-like defects in an otherwise perfect system. I do not believe this represents real world disorder and I think the authors have not fully considered the recent literature on the effect of fabrication disorder for exactly these structures, e.g., the previously mentioned references [C. Rosiek et al, Nature Photonics 17, 386 (2023)] and [E. Verhagen, Proceedings Volume PC12196, Active Photonic Platforms 2022, SPIE 2022; <https://doi.org/10.1117/12.2632855>]. If topological waveguides are not to be protected against fabrication disorder is the mechanism of the slow light in these waveguides enough to grant publication in Nature Communications? I also want to highlight that the authors responded to my comment on the effect of disorder by mentioning that the operating wavelength is three orders of magnitude longer than in the telecom-band valley-Hall waveguides of the references, but the effect of disorder would be the same if the disorder is of several microns as they themselves suggest. What matters is the size of disorder relative to the pitch and in the waveguides of the references that would also be three orders of magnitude smaller (few nanometers at maximum). This can be easily understood from perturbation theory expressions to the loss rates [see for example S. Hughes, L. Ramunno, Jeff F. Young, and J. E. Sipe, Phys. Rev. Lett. 94, 033903 (2005)]. In conclusion, what would make slow light in these waveguides appealing for technological applications in integrated photonics, i.e., their supposed topological protection, probably does not manifest in their structures. Maybe a discussion on the disorder levels of the structures (roughness) or loss measurements could be helpful.

Our response:

We thank the reviewer for highlighting the discussion on robustness. Here, we want to highlight that in this study, we aim to underscore our elucidation of the underlying mechanism governing the topological slow light effect in the bearded interface valley photonic crystal (VPC) waveguide. Additionally, we have shown the topological protection of slow light modes against point-like and non-Hermitian defects. The profound insights derived from our research are poised to significantly contribute to the comprehension of non-Hermitian physics and bear substantial technological importance for the development of highly efficient integrated circuits, as elaborated in detail below (Fig. R3).

Regarding the reviewer's concern towards the topological protection of slow light modes against fabrication disorder, we emphasize that to outline the loss characteristics of the topological slow light modes it requires a dedicated investigation, due to the following reasons:

- It requires the fabrication of a large volume of samples under identical fabrication conditions to have the structural disorder with practically equivalent statistics.
- The above cited work (Nature Photonics 17,386, 2023) employs the equation $\langle \ln T_{prop,L}(\lambda) \rangle = -\alpha(\lambda)L$, which assumes that the expectation value of log-transmittance through the VPC waveguide linearly depends on its propagation length (L). This assumption was also mentioned in the cited work (in the description following Eq. 1):

“Although such Beer–Lambert-like attenuation has been theoretically shown to fail for particular periodic monomode waveguides, the moderate values of n_g explored here and the state-of-the-art nanofabrication process justify the model.”

Our study, however, uncovers that in the bearded interface VPC waveguide, the Poynting vector of kink states undergoes a detour mediated by the magnetic phase vortices. Consequently, for extracting the propagation loss in the VPC waveguide under slow light conditions, there is likely a need to modify the analytical expression of the transmission model. As such, we believe this could be one of the reasons that in the above-cited works the authors have not found any advantages of topological mode over the trivial mode in the slow light regime.

Also, it is noteworthy that, in the cited work “Nature Photonics 17,386,2023”, while similar propagation loss between the topological and trivial modes is observed for group index within the range $26 < n_g < 28$, the topological mode exhibits a visibly lower loss for $n_g < 25$ with a lack of data for fair comparison in the range $n_g > 30$ (where we can observe the beginning of a slight decrease in the loss of topological mode again). It is plausible that the propagation loss of the topological mode follows a more linear trend with respect to the group index as opposed to the quadratic/exponential-like trend for the trivial mode. Since the experimentally estimated group index for our waveguide is ~ 50 , an extensive investigation is required for fair comparison. Also, the Beer-Lambert model is not suitable for waveguide with a high group index of 50 as pointed out previously.

To clarify, our current work unveils the underlying mechanism of topological slow light mode in VPC and demonstrates its passive and active control through bandgap engineering and by introducing non-Hermitian photo-defect, respectively. Nevertheless, to gain more comprehensive understanding on the robustness against fabrication disorder, additional extensive investigations are warranted, which we intend to explore in our subsequent research endeavours.

Furthermore, regarding the reviewer's concern about the practical application of our work, we highlight that the appearance of slow light provides important prospects for integrated photonics applications. Since for integrated photonics, device footprint plays a crucial role, and we have shown, in VPC waveguide slow light persists even at the presence of sharp bends. This attribute proves highly advantageous in reducing the footprint of on-chip devices. Furthermore, the slow light regime significantly enhances light-matter interactions, making it viable for on-chip signal processing and miniaturized sensors.

Below, we outline some of the specific applications that would highly benefit from the VPC slow light for the development of highly functional integrated photonic circuits: Fig. R3 depict the schematic of topological integrated photonic chip where the robustness of slow light is being harnessed to achieve highly efficient integrated photonic circuits.

1. **On-chip delay line:** The combination of slow light and its immunity towards sharp bends allow us to design a highly compact on-chip delay line (Fig. R3).
2. **High-speed terahertz modulator:** Achieving ultra-high-speed modulation of terahertz waves remains a challenging task for emerging 6G communication devices. Currently, researchers employ indirect methods, such as manipulating optical or electronic signal to modulate the terahertz signals. The advent of topological slow light modes offers opportunities to develop direct terahertz on chip modulator either by integrating quantum materials or devising Mach Zender type modulator, shown in Fig. R3.
3. **On-chip topological Mach Zender interferometer (MZI):** An on-chip Mach Zender interferometer holds potential utility in quantum metrology and highly sensitive phase and amplitude detectors. Leveraging topological slow light, an ultra-compact MZI can be devised, as highlighted in Fig. R3.
4. **Ultra-high-quality factor on-chip slow light cavity:** Devising a topological cavity with bearded interface domain takes the advantage of slow light and topological protection. This opens opportunities to create highly compact topological slow light cavities with unprecedented quality factor. Integrating the topological slow light cavity with waveguide enables several functionalities such as on-chip channel filtering, signal storage and sensing.

Fig. R3: A conceptual visualization of topological photonic integrated chip, where the robustness of slow light enables the realization of several functional components.

In addition to these main arguments, I would like to point a few more aspects:

R1.4: The authors have extended on these valuable simulations about the photoexcitation experiment. In relation to Fig. S13, the authors say: “it becomes apparent that the kink state takes a detour around the circular region and retain its forward and counter-propagating waves after traversing the circular region. In contrast, for the trivial mode, there is a significant scattering and back reflection with an increase in silicon conductivity”. I sort of understand why the authors say that the mode takes a detour, but I want to highlight that the fact that it retains the forward and counter-propagating waves after it is not very relevant. It is a single-moded waveguide so only that mode can potentially propagate after the defect (as also happens for the trivial mode). I think plots of simulated transmission as a function of the conductivity and experimental transmission as a function of the pump power are needed to see how efficiently the topological mode bypasses those defects. The same applies to the simulations of the

artificial disorder (Fig. S16). If the “active control” of the group delay incurs large losses, then it is not so interesting.

Our response:

We thank the reviewer for bringing up this point. In response to the reviewer’s suggestion, we have generated plots illustrating the transmission spectra as a function of conductivity in our simulations and optical pump power in our experiments, as depicted in Fig. R4a and R4b, respectively. Notably, the photoexcitation (or changes in conductivity) does not compromise the topological protection within the VPC waveguide. Consequently, it maintains a consistent flat and high transmission over the same bandwidth even when subjected to two sharp bends for varying pump power (or conductivity), as observed in Fig. R4a and R4b. However, it is important to emphasize that the broadband modulation observed with increase in optical pump (or changes in conductivity) is attributed to the influence of free carriers, which lead to the attenuation of terahertz waves.

Fig. R4: Transmission through Z-shaped slow light VPC waveguide (a) Simulated transmission spectra as a function of varying conductivity of silicon for the non-Hermitian defect region. **(b)** Experimentally recorded transmission spectra at various optical pump power. The triangular marker is where the group delays are extracted.

R1.5: Whenever the authors talk about topological protection, they should specify which topological waveguides they are talking about (time-reversal-symmetric or not) and against which type of defects.

Our response:

We thank the reviewer for pointing this out. In the revised manuscript, we have specified the type of topological waveguide and the nature of robustness.

The updated text in the revised manuscript appears at page 2, line 24 as:

“Notably, the robust transport in VPC relies fully on geometric symmetries. As a result of this, VPC waveguide with sharp corners including 60- or 120-degree bends which preserve the hexagonal lattice symmetry exhibits negligible bending loss.”

R1.6: The authors say: “Compared to conventional PC platforms, the vortex-driven slow light effect in VPCs offer significant flexibility to engineer the functionality of topological slow light photonic devices.” It is quite unclear to me why the authors say this and if it is justified by their work.

Our response:

We thank the reviewer for pointing out the ambiguity. We rephrased the sentence as:

“Compared to conventional PC platforms, the vortex-driven slow light effect in VPCs offer flexibility to control the slowness by engineering the interaction between forward and counter-propagating waves.”

In the revised manuscript, it is highlighted at page 3, line 14.

R1.7: Fig. 1a is still very poorly rendered. The arrows are extremely pixelized upon zooming.

Our response:

We thank the reviewer for the feedback. We will provide vector graphics to resolve this issue.

R1.8: Regarding the distinction between the topological and trivial mode, which has been tackled more in detail in the updated version of the manuscript, the authors say: "It is noteworthy that, to precisely distinguish between the topological and trivial modes within the bearded interface VPC waveguide, it is appropriate to observe the presence of counter-propagating waves (see detailed discussion in SI, section S1)." For me this is quite a tautological argument, given that this is what they are actually trying to demonstrate in the manuscript. I think another argument should be provided, like a simulation of the transmission across Z-shaped bends.

Our response:

We thank the reviewer for highlighting this point. Following the reviewer's suggestion, we have plotted the transmission through Z-shaped VPC waveguide. Fig. R5a and R5c show the projected band diagram for both Type I and Type II bearded interface with triangular air holes ($\nu = 3$). The solid and dashed lines highlight the topological and trivial modes, respectively. The topological and trivial modes are distinguished by examining the transmission spectra through Z-shaped Type I and Type II VPC waveguides, as shown in Fig. R5b and R5d. The high transmission observed within the frequency range (indicated by orange shaded region) corresponding to the topological edge states signify the topological nature of the lower band for Type I bearded interface.

Similarly, for Type I bearded interface with hexagon ($\nu = 6$) and nonagon ($\nu = 9$) air holes, we observe that the lower bands exhibit topological characteristics, as evident from the high transmission within the frequency range covered by the topological edge states, as shown in Fig. R5e, R5f, R5i and R5j.

In case of Type II bearded interface with hexagon and nonagon air holes, where the radii (r) of air holes are $r > \frac{a}{2\sqrt{3}}$, they lead to a physically separated photonic crystal with an air-slot waveguide at the bearded interface. This configuration is illustrated in a recent publication Opt. Express 30, 33035-33047 (2022). However, the presence of overlapping air holes (in the case of hexagon and nonagon) forming an asymmetric air-slot waveguide allows the backwards coupling of wave at the bends due to the propagation of the edge states along the domain wall. To address this issue, a composite bearded-zigzag interface junction was proposed in Opt. Express 30, 33035-33047 (2022).

Fig. R5g and R5k show the projected band diagram for the Type II bearded interface along with composite zigzag interface. The topological edge modes are highlighted by blue solid line, while the edge modes corresponding to zigzag interface are depicted by black solid line. In projected band diagram, the region covered exclusively by the topological edge modes is highlighted by blue shaded region.

Fig. R5h and R5l depict the transmission spectra through the composite interface with 60° bends (according to Opt. Express 30, 33035-33047 (2022)) constructed using hexagon and nonagon air holes, respectively. The high transmission observed within the frequency region highlighted by blue shaded region confirms the topological nature of the lower and upper bands.

Fig. R5. Identification of topological and trivial modes in VPC waveguide: (a, b) Projected band diagram and transmission spectrum for Type I bearded interface with triangular air holes ($\nu = 3$), respectively. (c, d) Projected band diagram and transmission spectrum for Type II bearded interface with triangular air holes ($\nu = 3$), respectively. (e, f) Projected band diagram and transmission spectrum for Type I bearded interface with hexagon air holes ($\nu = 6$), respectively. (g, h) Projected band diagram and transmission spectrum for Type II bearded interface with hexagon air holes ($\nu = 6$), respectively. (i, j) Projected band diagram and transmission spectrum for Type I bearded interface with nonagon air holes ($\nu = 9$), respectively. (k, l) Projected band diagram and transmission spectrum for Type II bearded interface with nonagon air holes ($\nu = 9$), respectively.

R1.9: I feel like Fig. 2 could now be in the SI, since Fig. 3 serves roughly the same purpose but gives much better arguments for the conclusion that the authors make.

Our response:

We thank the reviewer for the suggestion. However, we want to highlight that Fig. 2 conveys an important part of our work. It provides an outlook to engineer the strength of slow light effect in VPC waveguide via bandgap tuning that is not limited to specific air hole shapes.

R1.10: Page 12, Lines 13-18: I think the authors should drop this discussion about the group index. This distinction between the local variation and the global variation was brought up during the review and is now corrected. The justification of why one should evaluate GD instead of n_g feels unnecessary now and may only add confusion.

Our response:

We thank the reviewer for pointing this out. In the revised manuscript we have removed the discussion of group index (n_g) to avoid any confusion.

R1.11: In Table 1, the column with “Topological protection” is confusing. Do the authors imply that they have shown topological protection? This connects to my previous point number 2 here.

Our response:

We thank the reviewer for highlighting this. In the revised manuscript we have substituted the heading “Topological protection” with “Robust transport at sharp bends” to distinctly emphasize its distinction from other state-of-the-art platforms.

R1.12: The “active” in the title is confusing. There is nothing active about the slow light. I believe the only way the word active could come in would be in connection to the words group delay, e.g., “active group delay control” or something similar.

We thank the reviewer again. Following the reviewer’s suggestion, we have changed the title to avoid any confusion. The revised title is as follow:

“Slow Light Topological Photonics with Counter-Propagating Waves and its Active Control on a Chip”

Reviewer #2 (Remarks to the Author):

The authors have addressed all the concerns in my last report. I am happy to recommend it for publications in Nature Communications in its present form.

Our response:

We thank the reviewer for recommending our work for publication.

Reviewer #3 (Remarks to the Author):

R3.1: I appreciate the authors’ great efforts in responding to my comments. I am satisfied with their response except for the assignment of trivial and topological bands. The authors assigned the modes that have back-propagation modes as topological. Naively, it is expected that those modes show high transmittance over sharp bends. Before recommending the manuscript for publication, I request the authors to demonstrate this feature by showing the computed transmission spectra in the structures with sharp bends. (I suppose that the transmission spectra in Figs. S1. 3 and 4 are the ones for straight waveguides). It is informative to point out the one-to-one correspondence between the waveguiding properties against sharp turns, which is widely known as a feature of topological modes, and the presence of the counter-propagating waves.

Our response:

We thank the reviewer for the positive comments and recommending our work for publication. Below we address the reviewer comment in detail.

Regarding the assignment of trivial and topological bands we have plotted the transmission through Z-shaped VPC waveguide. Fig. R6a and R6c show the projected band diagram for both Type I and Type II bearded interface with triangular air holes ($\nu = 3$). The solid and dashed lines in the diagrams highlight the topological and trivial modes, respectively. The topological and trivial modes are distinguished by examining the transmission spectra through Z-shaped Type I and Type II VPC waveguides, as shown in Fig. R6b and R6d. The high transmission observed within the frequency range (indicated by orange shaded region) corresponding to the topological edge states signify the topological nature of the lower band for Type I bearded interface.

Similarly, for Type I bearded interface with hexagon ($\nu = 6$) and nonagon ($\nu = 9$) air holes, we observe that the lower bands exhibit topological characteristics, as evident from the high transmission within the frequency range covered by the topological edge states, as shown in Fig. R6e, R6f, R6i and R6j.

In case of Type II bearded interface with hexagon and nonagon air holes, where the radii (r) of air holes are $r > a/2\sqrt{3}$, they lead to a physically separated photonic crystal with an air-slot waveguide at the bearded interface. This configuration is illustrated in a recent publication Opt. Express 30, 33035-33047 (2022). However, the presence of overlapping air holes (in the case of hexagon and nonagon) forming an asymmetric air-slot waveguide allows the backwards coupling of wave at the bends due to the propagation of the edge states along the domain wall. To address this issue, a composite bearded-zigzag interface junction was proposed in Opt. Express 30, 33035-33047 (2022).

Fig. R6g and R6k show the projected band diagram for the Type II bearded interface along with composite zigzag interface. The topological edge modes are highlighted by blue solid line, while the edge modes corresponding to zigzag interface are depicted by black solid line. In projected band diagram, the region covered exclusively by the topological edge modes is highlighted by blue shaded region.

Fig. R6h and R6l depict the transmission spectra through the composite interface with 60° bends (according to Opt. Express 30, 33035-33047 (2022)) constructed using hexagon and nonagon air holes, respectively. The high transmission observed within the frequency region highlighted by blue shaded region confirms the topological nature of the lower and upper bands.

Fig. R6. Identification of topological and trivial modes in VPC waveguide: (a, b) Projected band diagram and transmission spectrum for Type I bearded interface with triangular air holes ($\nu = 3$), respectively. (c, d) Projected band diagram and transmission spectrum for Type II bearded interface with triangular air holes ($\nu = 3$), respectively. (e, f) Projected band diagram and transmission spectrum for Type I bearded interface with hexagon air holes ($\nu = 6$), respectively. (g, h) Projected band diagram and transmission spectrum for Type II bearded interface with hexagon air holes ($\nu = 6$), respectively. (i, j) Projected band diagram and transmission spectrum for Type I bearded interface with nonagon air holes ($\nu = 9$), respectively. (k, l) Projected band diagram and transmission spectrum for Type I bearded interface with nonagon air holes ($\nu = 9$), respectively.

Regarding the reviewer's concern on one-to-one correspondence between waveguiding properties at sharp turns and counter-propagating waves in VPC, we would like to emphasize that the robust transport in VPC waveguide primarily results from the presence of magnetic phase vortices, which are

present in both zigzag and bearded interfaces, Consequently, VPC waveguides with both zigzag and bearded interfaces exhibit excellent signal transmission even at sharp turns.

In contrast, the occurrence of counter-propagating waves necessitates two specific conditions: **(a)** the presence of magnetic phase vortices with non-zero Berry curvature and **(b)** the existence of glide symmetry. These requirements are exclusively satisfied by VPC waveguides with bearded interfaces. Notably, the presence of counter-propagating waves leads to a slow light effect in the VPC waveguide by introducing an additional detour for the valley kink states, as illustrated in Fig. R7. In conclusion, in bearded interface VPC waveguide, magnetic vortices facilitate robust transport around sharp bends while counter-propagating waves give rise to the intriguing slow light effect.

Furthermore, we have prepared a supplementary video SVc to address the one-to-one correspondence between the waveguiding and the presence of counter-propagating waves. This video demonstrates the evolution of the Poynting vector profile, transitioning from “only forward propagating waves” to “forward and counter-propagating waves” in the bearded interface VPC waveguide as we sweep across the frequencies corresponding to the K-valley to the band edge. As the magnetic phase vortices exist for wavevectors ranging from the projected K point ($k_x = 2\pi/3a$) to the band edge ($k_x = \pi/a$), we observe the appearance of counter-propagating waves from the frequency corresponding to the projected K point to the band edge frequency, as clearly evident in SVc.

To illustrate more clearly that counter-propagating waves sustain even at the presence of sharp corners, we plot the Poynting vector (Fig. R7a), corresponding to the frequency marked by square dot in the transmission spectra (Fig. R7b). It is evident that counter-propagating waves exist even traversing the sharp corners.

Fig. R7. Existence of counter-propagating waves at sharp bends (a) The Poynting vector corresponding to the frequency marked in the **(b)** transmission spectrum.

R3.2: In addition, it is very interesting that the topological band in the Type 1 structure changes from the lower-frequency band to the higher-frequency band when ν changes from 3 to 6 and 9. Could the authors discuss the reason for the change?

Our response:

We thank the reviewer for pointing out that the method of comparing transmission between straight waveguide and those with sharp bend is the appropriate method for identifying topological bands. We have now confirmed that the lower band is always the topological band (see Fig. R6).

REVIEWER COMMENTS

Reviewer #1 (Remarks to the Author):

The authors have addressed most of the previous points adequately, with considerable additional simulations to elucidate the correspondence between the topological/trivial denomination and the existence of counter-propagating wavefronts, as well as the specific role of the glide symmetry and the magnetic vortices in the observed phenomena. I can now recommend the manuscript for publication.

Reviewer #3 (Remarks to the Author):

I deeply appreciate the authors' additional simulations for identifying the origins of in-gap guided modes. However, I still have several comments.

1) I recommend the authors to provide additional explanations about the mode assignments.

A) The type II structure shown in Fig. 2 (g) is essentially same as the structure discussed in X.-T. He et al., Nat. Commun. Vol. 10, 872 (2019). In the report, the higher-frequency band, which is assigned as a trivial mode in the current manuscript, exhibits robust light-guiding properties over sharp turns. Please discuss the reasons for this discrepancy.

B) It is surprising to me that there are two topological bands in some structures. In the widely accepted discussion, the number of topological in-gap modes at the interface between two topologically distinct VPhCs is 1 since the valley Chern numbers are $1/2$ or $-1/2$ for the two VPhCs. So it is quite informative that the authors will elaborate on why two topological modes appear in these structures. Even just adding the relevant references is useful.

C) It is also interesting that the topological band, the lower-frequency band, in Type I in Fig. 2 (a) becomes a trivial band when the shape of holes changes from triangles to hexagons. Any physical or mathematical discussions explaining the transition are highly expected.

2) Regarding the mechanism of active tuning, as Reviewer 1 pointed out, the authors should elaborate on the mechanism and make the explanation consistent throughout the paper. In fact, I am still wondering whether the active tuning really relies on the adjustment of the coupling strength between the counter-propagating waves. As the authors explained in the abstract, the coupling strength depends on the spatial separation between the counter-rotating phase vortices. I could find no remarkable change in the distance in the simulation results (Fig. 4 (d) and 4 (e)) but simply weakened counter-

propagating waves. I agree that active tuning requires the existence of counter-propagating waves. But this requirement doesn't mean that the "coupling strength" is adjusted.

3) The authors also explained that "the kink state takes a detour around the circular region" on page 13. After reading the manuscript carefully again, this sounds counterintuitive at least for me. I think it will take a longer time for light to reach a certain point when the light detours, resulting in a larger group delay. I expect the authors to correct my probable misunderstanding.

4) (minor) (Page 2, LL24-27) Valley kink states are known to be robust as far as the intervalley scattering is well suppressed. 60-degree bends change the propagation direction from K to K' (vice versa). Are the kink states robust even against such bends exchanging the valley? It would be useful if the authors could cite some references that theoretically explain the robustness against 60-deg bends.

The following are the comments I have after reading the correspondence between Reviewer 1 and the authors.

5) (R1.2: example of square-lattice waveguides) A 90-deg bend for a C4-based glide-plane waveguide was examined in W. Dai et al., CLEO-PR2022, CFP81-03 (2022). A reasonably high transmittance over the corner was reported in Fig. 1(d). Related to 1)(A), see also Fig. 1 (a)(b) in the abstract.

6) (R1.3: robustness against the random fluctuation) I agree that the assessment at the frequencies with a large group index will be complicated and further studies are necessary. However, it is a fact that there is no inherent topological protection against random disorders since topological protection of valley kink states relies on structural symmetry. VPhC waveguides are robust against turns and defects with some specific types, which is advantageous for some applications. On the other hand, regarding the robustness for random fluctuations, VPhC waveguides would be only beneficial within a limited range of fluctuations. This is one of the important issues for considering the practical applications of time-reversal topological photonic systems. Given this, it seems fair to cite some relevant papers on the issue and discuss that the effect is less significant in THz than in telecommunication wavelength since the operation wavelength is long.

Response Letter to Reviewers

We would like to thank all the reviewers again for their invaluable comments which have helped us significantly improve the strength of our study as well as the clarity of our manuscript.

In the subsequent text, the comments from the reviewer are presented in 'black' text, followed by our comprehensive response highlighted in 'blue'. Taking into consideration the reviewer's comment, the revisions incorporated in the revised manuscript are highlighted in yellow. The modified texts in the revised manuscript are posted here in the *italic* font.

Reviewer #1 (Remarks to the Author):

The authors have addressed most of the previous points adequately, with considerable additional simulations to elucidate the correspondence between the topological/trivial denomination and the existence of counter-propagating wavefronts, as well as the specific role of the glide symmetry and the magnetic vortices in the observed phenomena. I can now recommend the manuscript for publication.

Our response:

We thank the reviewer for their constructive comments which helped strengthen our work. We would also like to thank the reviewer for recommending our manuscript for publication.

Reviewer #3 (Remarks to the Author):

I deeply appreciate the authors' additional simulations for identifying the origins of in-gap guided modes. However, I still have several comments.

I recommend the authors to provide additional explanations about the mode assignments.

R3.1a: The type II structure shown in Fi. 2 (g) is essentially same as the structure discussed in X.-T. He et al., Nat. Commun. Vol. 10, 872 (2019). In the report, the higher-frequency band, which is assigned as a trivial mode in the current manuscript, exhibits robust light-guiding properties over sharp turns. Please discuss the reasons for this discrepancy.

Our response:

We would like to thank the reviewer for pointing out the error in the identification of topological and trivial bands for Type II structure with triangular air holes. As a matter of fact, we closely followed the method in X.-T. He et al., Nat. Commun. Vol. 10, 872 (2019) for the purpose of identifying topological and trivial modes using a chiral point source for both Type I and Type II interfaces. We have now corrected the topological band for Fig. 2(a) corresponding to the Type II structure in Fig. 2(g) in the manuscript (see Fig. R1). We have also updated Fig. 2(a) of the manuscript and Fig. S1.2 of the supplementary information correspondingly.

Fig. R1. Identification of topological band by comparing the transmittance.

R3.1b: It is surprising to me that there are two topological bands in some structures. In the widely accepted discussion, the number of topological in-gap modes at the interface between two topologically distinct VPhCs is 1 since the valley Chern numbers are $1/2$ or $-1/2$ for the two VPhCs. So it is quite informative that the authors will elaborate on why two topological modes appear in these structures. Even just adding the relevant references is useful.

Our response:

We thank the reviewer for this comment. Indeed, the bulk-boundary correspondence dictates that the number of topological in-gap modes at the interface between two topologically distinct materials which is equal to the difference between their Chern number. However, this is not entirely true for photonic topological insulator, since the valley Chern number that is numerically computed from the eigen solutions of the Maxwell's eigenequations will only be $\pm 1/2$ when the spatial inversion symmetry is not broken (i.e., the Dirac cone is still present at K/K' valleys in the photonic band structure).

In the photonics community, there are several recent reports showing the deviation of valley Chern number from $\pm 1/2$. For instance, in Yang, J.-K. et al. Phys. Rev. Research 3, L022025 (2021), the authors have shown that the valley Chern number continuously vary from 0 to $\pm 1/2$ as the asymmetry between the two air holes in a valley photonic crystal unit cell change. In addition, we also numerically compute the valley Chern number for valley photonic crystals with triangular air holes, showing that the magnitude of valley Chern number decreases from $1/2$ as we break the spatial inversion symmetry, as shown in Fig. R2.

Fig. R2. Numerically computed valley Chern number as a function of air hole asymmetry for valley photonic crystals with triangular air holes.

Since the valley Chern numbers are not exactly half-integers in valley photonic crystals with band gap, the theoretical application of bulk-boundary correspondence may not be obvious.

On a side note, the structures with two topological bands in our manuscript are all air-slot-like Type II bearded interface waveguide (due to the air hole radius exceeding the apothem of triangular unit cells $r > 0.5a/\sqrt{3}$). Quantifying the number of topological modes in such cases likely requires a whole separate study.

R3.1c: It is also interesting that the topological band, the lower-frequency band, in Type I in Fig. 2 (a) becomes a trivial band when the shape of holes changes from triangles to hexagons. Any physical or mathematical discussions explaining the transition are highly expected.

Our response:

Thank you for the comment. However, based on Fig. 2 of our previously submitted version of the manuscript, the lower-frequency band of Type I interface remains topological even when the shape of air holes changes from triangle to hexagon as highlighted in Fig. R3 below.

Fig. R3. Fig. 2(a) and 2(b) of the previously submitted manuscript. Here the lower frequency band is topological for Type I structure with triangular and hexagonal holes, respectively.

In response to comment R3.1a above, Fig. 2(a) and Fig. 2(b) of the manuscript is now updated to show the correct topological identification for Type II interface as well (see Fig. R4 below).

Fig. R4. Corrected figure 2(a) and 2(b) of the manuscript.

R3.2: Regarding the mechanism of active tuning, as Reviewer 1 pointed out, the authors should elaborate on the mechanism and make the explanation consistent throughout the paper. In fact, I am still wondering whether the active tuning really relies on the adjustment of the coupling strength between the counter-propagating waves. As the authors explained in the abstract, the coupling strength depends on the spatial separation between the counter-rotating phase vortices. I could find no remarkable change in the distance in the simulation results (Fig. 4 (d) and 4 (e)) but simply weakened counter-propagating waves. I agree that active tuning requires the existence of counter-propagating waves. But this requirement doesn't mean that the "coupling strength" is adjusted.

Our response:

We thank the reviewer for pointing out this ambiguity. We have removed the claim of "active tuning via adjustment of coupling strength between the counter-propagating waves" and modified the explanation to make it consistent throughout the manuscript.

Updated text in the revised manuscript at page 1, line 23:

"We also dynamically control the group delay by introducing a non-Hermitian defect using photoexcitation to adjust the relative strength of the counter-propagating waves."

Updated text in the revised manuscript at page 12, line 14:

"It is worth noting that, akin to tuning the slowness of kink state through band gap engineering (as shown in Fig. 2), photoexcitation also modifies the relative strength of the counter-propagating waves by modulating the terahertz waves through free carriers."

Updated text in the revised manuscript at page 13, line 15:

"For $\sigma = 0$ S/m, the transport of kink state is accompanied by forward and counter-propagating waves (Fig. 4c) where the electromagnetic waves are trapped temporally at each vortex before propagating to the next (see Supplementary Video SVb), which is the physical mechanism of topological slow light. As the conductivity of Si increases, free carrier densities increase and imparts losses, causing the photoactive region to act as a non-Hermitian defect that attenuates mainly the backward propagating waves (see Fig. 4d where the strength of backward Poynting vectors in "blue" weakens after the non-Hermitian defect). The attenuation of backward propagating waves leaves only the forward components and the waves are not trapped temporally at each vortex anymore after the photoactive region, resulting in the

decrease of GD. When the backward Poynting vector component is completely attenuated, the GD will be at minimum as shown in Fig. 4d and the inset of Fig. 4b for $\sigma = 10 \text{ S/m}$. With further increase in the conductivity of Si, both forward and backward propagating waves are attenuated about the position slightly after the photoactive region and the kink state now takes a detour as shown in Fig. 4e. This additional detour causes the GD to increase again as evident in the inset of Fig. 4b for conductivity more than 10 S/m . However, the additional group delay arising from such detour is less than those caused by the temporal trapping of counter-propagating waves at each vortex, hence the GD for $\sigma > 10 \text{ S/m}$ still is lower than the unperturbed case where $\sigma = 0 \text{ S/m}$.”

Updated text in the revised manuscript at page 15, line 7:

“The interaction between forward and counter-propagating waves gives rise to the slow light effect in VPC waveguides, which can be actively and passively controlled by adjusting the relative magnitude of their Poynting vectors.”

R3.3: The authors also explained that “the kink state takes a detour around the circular region” on page 13. After reading the manuscript carefully again, this sounds counterintuitive at least for me. I think it will take a longer time for light to reach a certain point when the light detours, resulting in a larger group delay. I expect the authors to correct my probable misunderstanding.

Our response:

We thank the reviewer for bringing up the possibility of this misinterpretation. In fact, a larger group delay is observed in our experimental data for optical pump power larger than 40 mW where the “detour” starts to appear in the Poynting vector as shown in Fig. 4b of our manuscript.

To better explain the initial decrease in group delay and the subsequent increase, we have updated the explanation in the revised manuscript which can be found at page 13, line 15 as:

“For $\sigma = 0 \text{ S/m}$, the transport of kink state is accompanied by forward and counter-propagating waves (Fig. 4c) where the electromagnetic waves are trapped temporally at each vortex before propagating to the next (see Supplementary Video SVb), which is the physical mechanism of topological slow light. As the conductivity of Si increases, free carrier densities increase and imparts losses, causing the photoactive region to act as a non-Hermitian defect that attenuates mainly the backward propagating waves (see Fig. 4d where the strength of backward Poynting vectors in “blue” weakens after the non-Hermitian defect). The attenuation of backward propagating waves leaves only the forward components and the waves are not trapped temporally at each vortex anymore after the photoactive region, resulting in the decrease of GD. When the backward Poynting vector component is completely attenuated, the GD will be at minimum as shown in Fig. 4d and the inset of Fig. 4b for $\sigma = 10 \text{ S/m}$. With further increase in the conductivity of Si, both forward and backward propagating waves are attenuated about the position slightly after the photoactive region and the kink state now takes a detour as shown in Fig. 4e. This additional detour causes the GD to increase again as evident in the inset of Fig. 4b for conductivity more than 10 S/m . However, the additional group delay arising from such detour is less than those caused by the temporal trapping of counter-propagating waves at each vortex, hence the GD for $\sigma > 10 \text{ S/m}$ still is lower than the unperturbed case where $\sigma = 0 \text{ S/m}$.”

R3.4: (minor) (Page 2, LL24-27) Valley kink states are known to be robust as far as the intervalley scattering is well suppressed. 60-degree bends change the propagation direction

from K to K' (vice versa). Are the kink states robust even against such bends exchanging the valley? It would be useful if the authors could cite some references that theoretically explain the robustness against 60-deg bends.

Our response:

We thank the reviewer for the query. Kink states have been experimentally shown to be robust against 60-degree bends in Yang, Y. et al. Nat. Photonics 14, 446–451 (2020) which shows high transmission over multiple 60-degree bends, and in Kumar, A. et al. Advanced Materials 2202370 (2022) where a topological cavity with two 60-degree bends is demonstrated.

The robustness of kink states against 60-degree bends is theoretically explained in the Supplementary Information of Ma, T. & Shvets, G. New J. Phys. 18, 025012 (2016). Following the reviewer's suggestion, we have cited this work at page 3, line 1 of our manuscript.

The following are the comments I have after reading the correspondence between Reviewer 1 and the authors.

R3.5: (R1.2: example of square-lattice waveguides) A 90-deg bend for a C_4 -based glide-plane waveguide was examined in W. Dai et al., CLEO-PR2022, CFP81-03 (2022). A reasonably high transmittance over the corner was reported in Fig. 1(d). Related to 1)(A), see also Fig. 1 (a)(b) in the abstract.

Our response:

Thank you for pointing out that 90-degree bends with glide symmetry in C_4 PhC is possible by creating the defect waveguide parallel to the ΓX direction (rotating the bulk by 45 degrees).

Nonetheless, we have shown a case where a glide symmetric waveguide does not show robust counter-propagating waves using a C_4 PhC with zero Berry curvature, in response to Reviewer #1 comment: "Can the authors show a glide plane symmetric waveguide that does not exhibit (counter-propagating waves) and is not topological?"

R3.6: (R1.3: robustness against the random fluctuation) I agree that the assessment at the frequencies with a large group index will be complicated and further studies are necessary. However, it is a fact that there is no inherent topological protection against random disorders since topological protection of valley kink states relies on structural symmetry. VPhC waveguides are robust against turns and defects with some specific types, which is advantageous for some applications. On the other hand, regarding the robustness for random fluctuations, VPhC waveguides would be only beneficial within a limited range of fluctuations. This is one of the important issues for considering the practical applications of time-reversal topological photonic systems. Given this, it seems fair to cite some relevant papers on the issue and discuss that the effect is less significant in THz than in telecommunication wavelength since the operation wavelength is long.

Our response:

We thank the reviewer for the well-thought comment. We agree with the reviewer that in THz frequency the effect of structural disorders over the transport of valley kink states would be less as compared to telecommunication wavelength due to long wavelength of THz waves. However, quantifying the robustness of valley kink states against the random structural fluctuations and disorders, particularly in THz frequencies is still an active area of research.

Following the reviewer's suggestion, we have cited the following papers in page 2, line 1 of our manuscript:

"However, quantifying the topological protection of transport of light in VPC is still an active area of research¹⁸⁻²¹."

18. Rosiek, C. A. et al. Observation of strong backscattering in valley-Hall photonic topological interface modes. *Nat. Photon.* 17, 386–392 (2023).
19. Arregui, G., Gomis-Bresco, J., Sotomayor-Torres, C. M. & Garcia, P. D. Quantifying the Robustness of Topological Slow Light. *Phys. Rev. Lett.* 126, 027403 (2021).
20. Hughes, S., Ramunno, L., Young, J. F. & Sipe, J. E. Extrinsic Optical Scattering Loss in Photonic Crystal Waveguides: Role of Fabrication Disorder and Photon Group Velocity. *Phys. Rev. Lett.* 94, 033903 (2005).
21. Mazoyer, S., Hugonin, J. P. & Lalanne, P. Disorder-Induced Multiple Scattering in Photonic-Crystal Waveguides. *Phys. Rev. Lett.* 103, 063903 (2009).

REVIEWERS' COMMENTS

Reviewer #3 (Remarks to the Author):

I am satisfied with the authors' responses and the corresponding revisions.

I have no further comment.